# Isoprene chemistry under upper-tropospheric conditions

Isoprene ($C_5H_8$) is the non-methane hydrocarbon with the highest emissions to the atmosphere. It is mainly produced by vegetation, especially broad-leaved trees, and efficiently transported to the upper troposphere in deep convective clouds, where it is mixed with lightning $NO_x$. Isoprene oxidation products drive rapid formation and growth of new particles in the tropical upper troposphere. However, isoprene oxidation pathways at low temperatures are not well understood. Here, in experiments at the CERN CLOUD chamber at 223 K and 243 K, we find that isoprene oxygenated organic molecules (IP-OOM) all involve two successive $OH^\bullet$ oxidations. However, depending on the ambient concentrations of the termination radicals ($HO_2^\bullet$, $NO^\bullet$, and $NO_2^\bullet$), vastly-different IP-OOM emerge, comprising compounds with zero, one or two nitrogen atoms. Our findings indicate high IP-OOM production rates for the tropical upper troposphere, mainly resulting in nitrate IP-OOM but with an increasing non-nitrate fraction around midday, in close agreement with aircraft observations.

Isoprene ($C_5H_8$) accounts for about half of the non-methane biogenic volatile organic compounds (VOC) emitted into the atmosphere, with an estimated annual rate of around 535 Tg year[−1][1]. Isoprene is highly reactive and can be oxidised by hydroxyl radicals ($OH^\bullet$), ozone ($O_3$) and nitrate radicals ($NO_3^\bullet$) to generate semi-volatile and low-volatile organic products that contribute to secondary organic aerosols (SOA). In regions with high isoprene emissions, its contribution to total SOA can exceed 55%[2]. The primary source of isoprene emission is broad-leaved plants, particularly those in tropical rainforests. However, significant emissions also occur in many other regions, marking isoprene as a globally important source of VOC[3,4]. Comprehensive reviews of isoprene oxidation chemistry and its contribution to aerosol growth under warm boundary layer conditions are provided by Wennberg et al.[5] and Carlton et al.[6]. However, little is experimentally known about isoprene oxidation chemistry at the cold temperatures of the upper troposphere.

Isoprene, a diene with two carbon–carbon double bonds (C=C), can undergo two rapid $OH^\bullet$ oxidations. Under warm boundary-layer conditions, most isoprene oxidation products are too volatile to drive aerosol formation or growth, as the small carbon backbone has fewer locations for intermolecular interactions, thus, limiting condensation. At boundary layer temperatures, first-generation products, such as isoprene hydroxy hydroperoxide (ISOPOOH, C* $\sim 5.6 \times 10^4\,\mu g\,m^{-3}$)[5,7,8], have volatilities within the intermediate volatile organic compounds (IVOC) range $\sim 10^2 - 10^6\,\mu g\,m^{-3}$[9]. In contrast, second-generation products fall into the range of semi-volatile organic compounds (SVOC) with volatilities $\sim 10^{-1} - 10^2\,\mu g\,m^{-3}$, allowing them to equilibrate with small aerosols and condense[9]. Hence, under boundary-layer temperatures, secondary oxidation, autoxidation and $C_{10}$ accretion product formation via $RO_2^\bullet$ cross-reactions are essential for isoprene to contribute to particle growth[10–12], despite the relatively low yields of secondary oxidation products and the limited efficiency of SVOC condensation. Beyond direct condensation, some isoprene oxidation products, particularly isoprene epoxy diol (IEPOX, C* $\sim 1.7 \times 10^4\,\mu g\,m^{-3}$[13]), an isomer of ISOPOOH (see SI section A), contribute to SOA formation through reactive uptake[10]. This occurs via heterogeneous chemistry between IEPOX and acidified sulphate seed particles[14–16]. Under such conditions, surface uptake of isoprene-generated glyoxal, methylglyoxal and IEPOX accounts for approximately 45% of the total SOA[2,17].

However, isoprene alone is not believed to contribute significantly to new particle formation (NPF) at atmospheric boundary-layer temperatures[18,19]. In contrast, isoprene-derived $C_5$ peroxy

✉e-mail: russell@iau.uni-frankfurt.de; curtius@iau.uni-frankfurt.de

radicals, $RO_2^{\bullet}$, can reduce the nucleation capability of the α-pinene system via interference with $C_{20}$ dimer formation, the main α-pinene oxidation products driving nucleation. In place of $C_{20}$ formation, more volatile $C_{15}$ dimers are formed[20,21]. The resulting $C_{15}$ compounds cannot nucleate as efficiently as the $C_{20}$ compounds, yet still contributes to particle growth at boundary layer temperatures[22].

Recent studies[23–26], show that isoprene can survive deep convective transport to reach the upper troposphere (8–12 km). During outflow from convective systems, high solar flux creates elevated concentrations of $HO_x$ ($OH^{\bullet} + HO_2^{\bullet}$) alongside lightning-generated $NO_x$ ($NO^{\bullet} + NO_2^{\bullet}$), with daytime values of $OH^{\bullet} \sim 10^6$ cm$^{-3}$, $HO_2^{\bullet} \sim 10^7$ cm$^{-3}$ and $NO^{\bullet} \sim 10^9$ cm$^{-3}$[25,27], creating an isoprene oxidation system which can nucleate and generating a large source of particles in a pristine environment[25,28]. Highly oxygenated compounds such as $C_5H_{12}O_6$, $C_5H_{11}NO_7$ and $C_5H_{10}N_2O_8$ appear to drive nucleation and growth under upper-tropospheric conditions. Shen and Russell et al.[28] found that isoprene oxidation products can nucleate under many different $NO_x$ conditions, and the addition of trace amounts of inorganic acids can enhance particle formation rates up to 100-fold. At such low temperatures, isoprene oxidation products, which are semi-volatile in the boundary layer, have decreased in volatility enough to contribute to new particle formation. Due to a low condensation sink, particles formed under such conditions could survive for several days and be transported hundreds of kilometres, contributing to a band of

particles observed at high altitudes over the tropics[29]. These recent findings underline the need for a comprehensive understanding of isoprene chemistry at low temperatures.

Under upper-tropospheric daytime conditions, $OH^{\bullet}$ radicals are the main oxidant of isoprene, other oxidants include $O_3$ and $NO_3^{\bullet}$, however, these reactions are slower in comparison. $OH^{\bullet}$ radicals attack one of the carbon–carbon double bonds on isoprene to form carbon-centred radicals, which in turn rapidly form peroxy radicals, $RO_2^{\bullet}$ via $O_2$ addition. The fate of $RO_2^{\bullet}$ involves a combination of unimolecular (autoxidation and isomerisation) and bimolecular reactions which can either propagate the radical chain to form another radical ($R'O_2^{\bullet}/RO^{\bullet}$) or terminate the radical chain to form closed shell species. Common termination radicals include $HO_2^{\bullet}$, $RO_2^{\bullet}$, $NO^{\bullet}$, $NO_2^{\bullet}$. Bimolecular termination of isoprene peroxy radicals forms three first-generation products depending on the terminating radical (Fig. 1): ISOPOOH ($C_5H_{10}O_3$), isoprene hydroxy nitrate (IHN, $C_5H_9NO_4$), and isoprene hydroxy peroxynitrate (IHPN, $C_5H_9NO_5$) for $HO_2^{\bullet}$, $NO^{\bullet}$ and $NO_2^{\bullet}$ termination, respectively. Each retains a double C=C, thus allowing it to undergo another rapid $OH^{\bullet}$ oxidation, see SI section A. Detailed chemical reaction schemes have been thoroughly investigated in laboratory flow tube and chamber experiments at room temperatures in previous studies[5,30–32]. However, the behaviour of isoprene and its oxidation products in the cold upper troposphere remained unknown until now. Due to the complexity of the $RO_2^{\bullet}$ chemistry, tailored

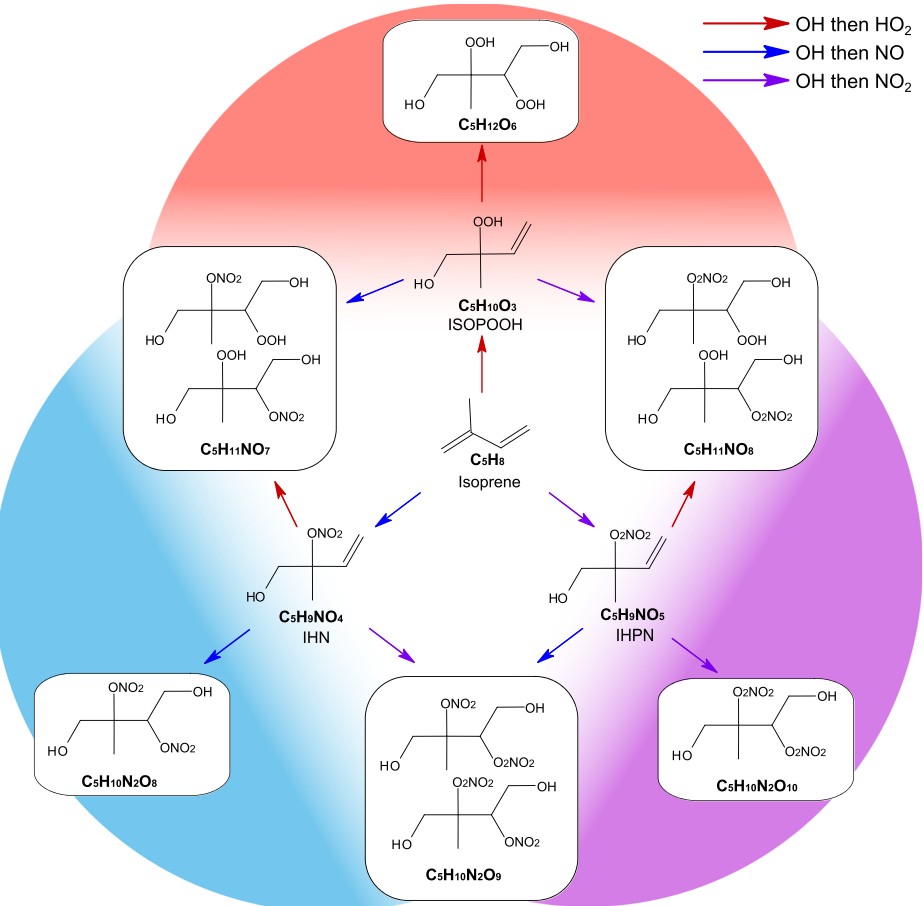

**Fig. 1 | Schematic of OH• initiated isoprene oxidation under daytime upper-tropospheric conditions.** Isoprene is oxidised by OH• to form a peroxy radical, ISOPOO• ($C_5H_9O_3$), which can be terminated by $HO_2^{\bullet}$, NO• or $NO_2^{\bullet}$ to form first-generation intermediates ISOPOOH (isoprene hydroxy hydroperoxide, $C_5H_{10}O_3$), IHN (isoprene hydroxy nitrate, $C_5H_9NO_4$), and IHPN (isoprene hydroxy peroxy nitrate, $C_5H_9NO_5$), respectively. The isoprene backbone contains two C=C bonds, therefore, the first-generation intermediates react with OH• to form peroxy radicals, which are terminated by $HO_2^{\bullet}$, NO• or $NO_2^{\bullet}$ to produce six distinct second-generation species ($C_5H_{12}O_6$, $C_5H_{11}NO_8$, $C_5H_{10}N_2O_{10}$, $C_5H_{10}N_2O_9$, $C_5H_{10}N_2O_8$ and $C_5H_{11}NO_7$), each with hydroperoxide groups, nitrate groups and peroxynitrate groups depending on which radicals led to termination. The red, blue and purple arrows represent, respectively, OH• then $HO_2^{\bullet}$, OH• then NO•, and OH• then $NO_2^{\bullet}$ reactions. This mechanism augments existing isoprene mechanisms to include peroxynitrates, a potentially important upper-tropospheric pathway. There are additional isomers that exist beyond those shown, and they follow similar chemical pathways.

experiments that covered upper-tropospheric relevant conditions are required to understand the chemical distribution.

In all isoprene oxidation mechanisms, methyl-vinyl ketone (MVK) and methacrolein (MACR) ($C_4H_6O$ isomers) are major products of isoprene oxidation[5,33]. However, even for upper-tropospheric temperatures, most $C_4$ products are too volatile to contribute to nucleation or even particle growth. For this reason, we will focus on the more highly oxygenated (predominantly $C_5$) compounds that drive new particle formation and use the same definition of Isoprene Oxygenated Organic Molecules (IP-OOM) as Shen and Russell et al.[28], molecules satisfying the following conditions: $n_C > 3$ and $n_O^{eff} > 3$[28], where $n_O^{eff} = n_O - 2 \bullet n_N$ as an $-ONO_2$ reduces the volatility of a compound roughly the same amount as an $-OH$ group[7,34]. We also use the corresponding IP-OOM subsets: $IP_{0N}$ for IP-OOM with $n_N = 0$, $IP_{1N}$ with $n_N = 1$, and $IP_{2N}$ with $n_N = 2$. This definition of IP-OOM removes MVK and MACR as well as key intermediates: ISOPOOH, IHN and IHPN—focusing on the oxidation products of these intermediates. Figure 4(a, b) in Shen and Russell et al.[28], indicates that all so-defined IP-OOM contribute to growth, regardless of their nitrogen composition. There are many possible chemical pathways of isoprene oxidation, leading to a vast number of species, but due to the importance of second-generation products observed in new particle formation[25,28], our study will focus on the simplified reaction scheme shown in Fig. 1.

We present a study of OH$^\bullet$-initiated isoprene gas-phase chemistry at low temperatures, 223 K (−50 °C) and 243 K (−30 °C), under different $NO_x$ conditions. Experiments were carried out in the CERN Cosmic Leaving OUtdoor Droplets (CLOUD) chamber during the CLOUD16 campaign (September–December 2023). Our study explores how the IP-OOM chemical composition changes under atmospheric concentrations, according to different levels of hydroperoxyl (HO$_2^\bullet$), nitric oxide (NO$^\bullet$) and nitrogen dioxide (NO$_2^\bullet$) radicals. We validate the results of our laboratory study through a comparison with aircraft measurements and expand the steady-state chemistry investigation where ambient observations cannot due to low residence times and lack of instrumentation. Ultimately, we assess the atmospheric importance of the reaction scheme using global model simulations.

## Results and discussion
### Peroxynitrate formation
Despite the presence of $O_3$ and $NO_3^\bullet$ during these experiments, isoprene reactivity and lifetime to oxidants are dominated by OH$^\bullet$ oxidation as shown in Fig. S1 (a, b).

Reaction of the first-generation isoprene peroxy radical with HO$_2^\bullet$ and NO$^\bullet$ forms a hydroperoxide (−OOH, ISOPOOH) and a nitrate (−ONO$_2$, IHN), respectively. Notably, the reaction of RO$_2^\bullet$ + NO$^\bullet$ can branch between nitrate and alkoxy radical formation (RO$^\bullet$), see SI section A. The branching ratio of this reaction, represented by $\alpha$, is complicated since it depends on many different factors, including chemical structure, pressure and temperature[35]. Peroxy radicals can also react with NO$_2^\bullet$ to form a peroxynitrate (−O$_2$NO$_2$, IHPN), although peroxynitrate formation is usually disregarded due to the rapid thermal decomposition back to RO$_2^\bullet$ + NO$_2^\bullet$.

$$RO_2^\bullet + HO_2^\bullet \xrightarrow{k_{HO_2}} ROOH + O_2 \tag{1}$$

$$RO_2^\bullet + NO^\bullet \xrightarrow{k_{NO}} RONO_2 \tag{2}$$

$$RO_2^\bullet + NO^\bullet \xrightarrow{k_{NO}} RO^\bullet + NO_2^\bullet \tag{3}$$

$$RO_2^\bullet + NO_2^\bullet \overset{k_{NO_2}}{\rightleftharpoons} ROONO_2 \tag{4}$$

ISOPOOH, IHN and IHPN can all undergo another OH$^\bullet$ oxidation to form more highly oxidised peroxy radicals and subsequently be terminated by HO$_2^\bullet$, NO$^\bullet$ and NO$_2^\bullet$ to form six distinct second-generation products as laid out in Fig. 1. Oxidation of these first-generation products is the main formation pathway of IP-OOM. IEPOX, another product from the oxidation of ISOPOOH[36], can also undergo a relatively fast OH$^\bullet$ oxidation, therefore, also contributes to IP-OOM formation.

Based on measurements under upper-tropospheric conditions, we propose that existing isoprene mechanisms should include per-oxynitrate (RO$_2$NO$_2$) formation, see Fig. 1. The only peroxynitrates commonly considered in Wennberg et al.[5] are peroxyacetyl nitrates (PANs); these have sufficiently long lifetimes for PANs to be an important $NO_x$ reservoir[37]. For lower tropospheric conditions, the non-PAN RO$_2^\bullet$ + NO$_2^\bullet$ ⇌ RO$_2$NO$_2$ reaction pathway for isoprene is not considered important and often neglected due to the rapid reverse reaction (thermal decomposition), making isolation of RO$_2$NO$_2$ very uncommon. However, at low temperatures, such as 223 K, thermal decomposition slows down and these compounds must be considered. For example, based on Zabel et al.[38], at $T = 223$ K and 800 mbar for a five-carbon alkyl chain, the decomposition lifetime is approximately $10^4$ s, compared to 0.5 s at 298 K, and would be expected to be even longer for lower atmospheric pressures. Consistent with this, peroxynitrates (such as $C_5H_9NO_5$, $C_5H_{10}N_2O_{10}$, $C_5H_{11}NO_8$, $C_4H_7NO_6$) were measured in experiments conducted at 223 K in the CLOUD chamber, see Fig. S2. In an isoprene oxidation experiment (Fig. S2) with high NO$_2^\bullet$:NO$^\bullet$, we observe an increase in both $C_5H_{10}O_3$ (ISO-POOH) and $C_5H_9NO_5$ (IHPN) with OH$^\bullet$ but not $C_5H_9NO_4$ (IHN), a commonly observed isoprene nitrate. Similar behaviour with $C_5H_{12}O_6$ and $C_5H_{10}N_2O_{10}$ compared to $C_5H_{10}N_2O_8$ occurs. Formation of nitrogen-containing compounds $C_5H_9NO_5$ and $C_5H_{10}N_2O_{10}$, without formation of isoprene nitrates $C_5H_9NO_4$ and $C_5H_{10}N_2O_8$ suggests these nitrogen-containing species differ from traditional RONO$_2$ chemistry. Literature values for the reaction rate coefficient of RO$_2^\bullet$ + NO$_2^\bullet$ are lacking (see SI section B and Table S1). Therefore, here we use our measurements together with kinetic arguments, as described in SI section B and Fig. S3, to estimate $k_{RO_2 + NO_2} = 3.4 \times 10^{-12}$ cm$^3$ s$^{-1}$. Our results indicate $k_{NO_2}$:$k_{NO} \sim 0.26$ for the low temperature isoprene system, which agrees with the literature of smaller peroxy radicals (Table S1) that $k_{NO_2}$:$k_{NO}$ lies between 0.1 and 0.5 for a wide range of RO$_2^\bullet$.

Using reaction rate coefficients from Table S2, Fig. S2a shows that the RO$_2^\bullet$ + NO$_2^\bullet$ pathway is comparable to the RO$_2^\bullet$ + HO$_2^\bullet$ pathway with negligible contribution from RO$_2^\bullet$ + NO$^\bullet$. This indicates that the reaction between NO$_2^\bullet$ and RO$_2^\bullet$ plays a critical role in determining the fate of RO$_2^\bullet$ at low temperatures. Thus, in Fig. 1, we have extended the reaction scheme presented in Curtius et al.[25] and Shen and Russell et al.[28] to include RO$_2$NO$_2$ chemistry, an important branch in the upper troposphere.

### IP-OOM production under different OH and $NO_x$ conditions
As discussed in the previous section, IP-OOM are the oxidation products of first-generation intermediates (ISOPOOH, IHN and IHPN) and IEPOX oxidation. IP-OOM production is influenced by multiple factors; here, we focus on the impact of OH$^\bullet$ and $NO_x$ by conducting experiments in which their concentrations are varied. Our results, at 223 K, show that across a wide range of OH$^\bullet$ concentrations ($5-75 \times 10^6$ cm$^{-3}$), 70% of the IP-OOM concentration retains their five-carbon backbone. Figure 2a depicts the $C_5$ IP-OOM distribution, categorising species into three groups based on their oxidation state. The first group (Saturated) consists of compounds that undergo two sequential OH$^\bullet$ oxidation and corresponding radical termination reactions, with an average H:C of 2.19. These species are typically saturated closed-shell structures with hydroxy (−OH), hydroperoxide (−OOH), nitrate (−ONO$_2$), or peroxynitrate (−OONO$_2$) functionality.

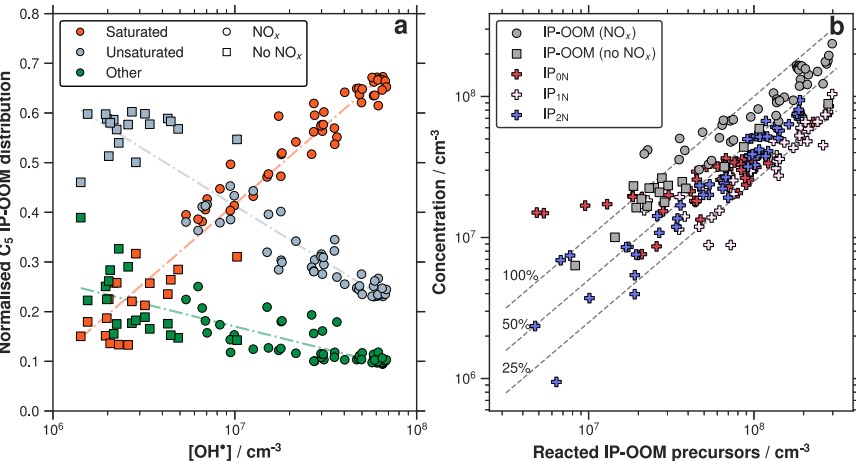

**Fig. 2 | Composition and yield of IP-OOM at 223 K. a** Normalised fraction of $C_5$ IP-OOM ($n_C = 5$, $n_O^{eff} > 3$) based on differing oxidation levels versus OH$^\bullet$ concentration. Isoprene is a diene, thus two OH$^\bullet$ additions can occur to form saturated products with −OH, −OOH, −ONO$_2$ and −O$_2$NO$_2$ functionality. Within this oxidation process, unsaturated products are also produced. These compounds have either retained a C=C bond, isomerised to contain a carbonyl group, C=O, or formed an epoxide, C-O-C. Saturation levels are determined by the hydrogen and nitrogen content: Saturated: $n_N = 0$, $n_H = 12$; $n_N = 1$, $n_H = 11$; $n_N = 2$, $n_H = 10$. Unsaturated: $n_N = 0$, $n_H = 10$; $n_N = 1$, $n_H = 9$. Other refers to the remaining $C_5$ IP-OOM, which includes radicals, H-abstraction products and products from nitrate and ozone oxidation. Circles and squares represent runs with and without NO$_x$, respectively.

The orange dashed line fits the data in the form Saturated fraction = 0.32 × log[OH$^\bullet$] − 1.81, the other fits are listed in Table S3. **b** total measured IP-OOM (grey) with (circles) and without NO$_x$ (squares) and IP$_{0N}$ (red cross), IP$_{1N}$ (pink cross) and IP$_{2N}$ (blue cross) versus reacted IP-OOM precursors, see SI section C for individual $x$-axis definitions. IP-OOM: $n_C > 3$, $n_O^{eff} > 3$; IP$_{0N}$, IP$_{1N}$ and IP$_{2N}$ are all subsets of IP-OOM, where $n_N = 0$, $n_N = 1$ and $n_N = 2$, respectively. The dashed lines indicate that the yield of IP-OOM is between 50 and 100%, and between 25 and 50% for each IP$_{0,1,2N}$. Using linear regression, the average yield of IP-OOM is 68%. For the individual subsets, specific IP-OOM precursors can be used to determine more accurate yields as described in SI section C. Using these definitions, the average yields are 36%, 31%, 42% for IP$_{0N}$, IP$_{1N}$ and IP$_{2N}$, respectively.

The second group (Unsaturated) with an average H:C of 1.89, includes a combination of first and second generation unsaturated molecules which contain either a carbon−carbon double bond (C=C), a carbonyl bond (C=O), or an epoxide group−products of IEPOX oxidation. The third group (Other), average H:C of 1.73, includes radicals, additional IEPOX oxidation products, and H-abstraction products. As shown in Fig. 2a, the $C_5$ IP-OOM fraction of saturated compounds increases with OH$^\bullet$, following a relationship of 0.32 × log[OH$^\bullet$] − 1.81 (Table S3). This trend persists across experiments with and without NO$_x$, highlighting the rate of the second OH$^\bullet$ addition, and underlining the necessity of two generations of OH$^\bullet$ oxidation for the formation of these products. In contrast, the $C_5$ IP-OOM fraction of unsaturated compounds decreases despite the concentration of this group and the total IP-OOM concentration increasing with OH$^\bullet$ concentration. Notably, the saturated group becomes dominant at OH$^\bullet$ concentrations of approximately $10^7$ cm$^{-3}$ (0.3 ppt), a concentration that has been detected at these temperatures (0.3 ppt at 213 K and 187 mbar $\sim$ $2 \times 10^6$ cm$^{-3}$)[25].

In Fig. 2b, we present the concentration of IP-OOM, IP$_{0N}$, IP$_{1N}$, and IP$_{2N}$ from the oxidation of IP-OOM precursors (ISOPOOH, IEPOX, IHN, and IHPN). Isomeric ISOPOOH and IEPOX are separated using chemical model calculations[28], see SI section A for details. Using a least-squares linear regression approach, we find that the average yield of IP-OOM from IP-OOM precursors, as described in SI section C, remains consistent regardless of the presence of NO$_x$, with a yield of approximately 68%. We show that the IP-OOM definition used in Shen and Russell et al.[28] effectively captures the majority of oxidation products from these precursors. For the individual groups, we have more information regarding their identity and we can therefore predict more accurate yields from the possible IP-OOM precursors (a detailed explanation of the yield calculations and $x$-axis definition for each subset can be found in SI section C). The average resultant yields are 36%, 31%, and 42% for IP$_{0N}$, IP$_{1N}$, and IP$_{2N}$, respectively; however, individual yields depend strongly on the specific radical ratios present. Given that the full mechanism of isoprene oxidation under low-

temperature conditions remains incomplete, our estimated average yields offer a preliminary basis for global and regional models to approximate IP-OOM production and assess its impact on new particle formation at a global scale.

## Case study: disentangling HO$_2$, NO and NO$_2$ termination

We distinguish between the different radical pathway contributions to IP-OOM, using three isolated cases: an HO$_2$ case, an NO case and an NO$_2$ case. The HO$_2$ case was carried out without NO$_x$ in the chamber, so we can characterise the role of HO$_2^\bullet$ and disentangle this from the NO$_x$ reactions. Unlike Teflon chambers, CLOUD is constructed of stainless steel and all wetted surfaces in or upstream of the chamber are metal. Therefore, the cleaned chamber is rigorously NO$_x$-free[39]. Building complexity on top of the HO$_2$ case, we injected NO$_x$−due to experimental constraints we cannot inject NO$^\bullet$ without producing NO$_2^\bullet$ and instead must control NO$_2^\bullet$:NO$^\bullet$ with differing light intensities. As a result, we have chosen two NO$_x$ cases: one with a ratio of NO$_2^\bullet$:NO$^\bullet$ ~100 (NO$_2$ case) and one with NO$_2^\bullet$:NO$^\bullet$ ~1 (NO case).

Each case contained similar OH$^\bullet$ levels at the same temperature and relative humidity, yet with vastly different radical ratios and concentrations (tabulated in Table 1).

### Table 1 | Experimental conditions for HO$_2$, NO, NO$_2$ cases in cm$^{-3}$

| Species | HO$_2$ case | NO case | NO$_2$ case |
|---|---|---|---|
| Isoprene[a] | $1.9 \times 10^{10}$ | $9.4 \times 10^9$ | $9.4 \times 10^9$ |
| OH$^\bullet$ | $1.0 \times 10^7$ | $1.3 \times 10^7$ | $1.3 \times 10^7$ |
| HO$_2^\bullet$ | $2.9 \times 10^8$ | $7.9 \times 10^7$ | $2.5 \times 10^8$ |
| NO$^\bullet$ | – | $2.8 \times 10^{10}$ | $6.2 \times 10^7$ |
| NO$_2^\bullet$ | – | $3.6 \times 10^{10}$ | $9.8 \times 10^9$ |

No NO$_x$ was present in the chamber during the HO$_2$ case; therefore, NO$^\bullet$ and NO$_2^\bullet$ values are below the limit of detection.
[a]Expected injected isoprene concentrations based on MFC settings.

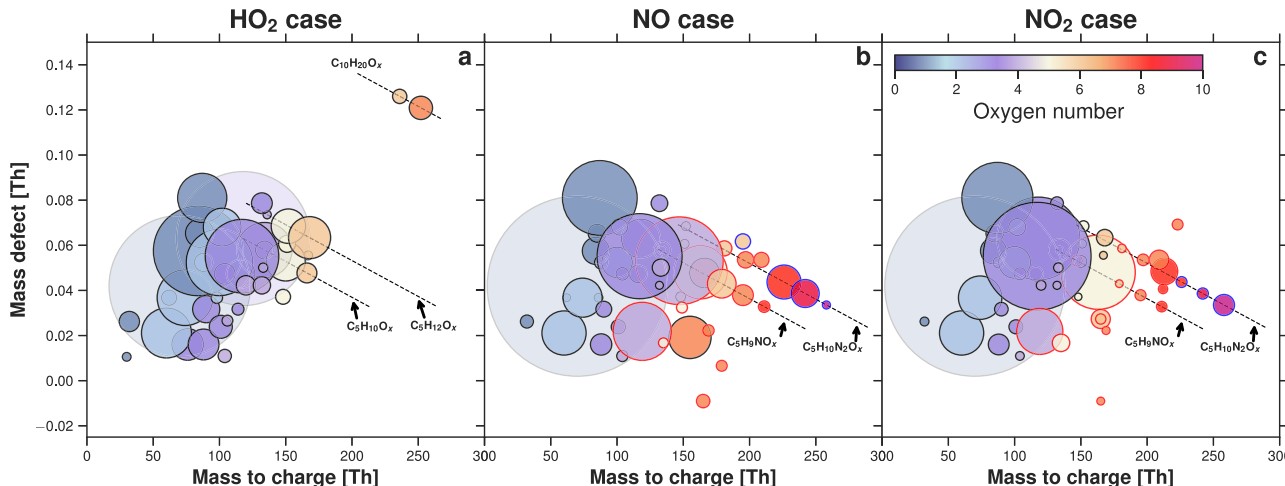

**Fig. 3 | Molecular composition for different radical dominant cases.** Mass defect (difference from integer mass) versus mass-to-charge for three radical distribution cases: **a** $HO_2$ case, **b** NO case and **c** $NO_2$ case. The data points are coloured by the number of oxygen atoms in the molecule and the outlines are coloured by the number of nitrogen present: $N = 0 \rightarrow$ black, $N = 1 \rightarrow$ red, $N = 2 \rightarrow$ blue. The symbol area is proportional to the log of the normalised species concentration. For concentrations $\geq 10^8$ cm$^{-3}$, symbols are faded to the background ($C_4H_6O$ for all cases and $C_5H_{10}O_3$ for $HO_2$ case). Only concentrations above $5 \times 10^5$ cm$^{-3}$ are shown in accordance with the highest limit of detection. This is a combination of three mass spectrometers applying $NO_3^-$, $Br^-$ and $NH_4^+$ ionisation methods.

Each individual case in Fig. 3 has a distinct IP-OOM distribution, highlighting the change from –OOH, –ONO₂ and –O₂NO₂ functionality. The faded circles represent $C_4H_6O$ (MVK / MACR) and $C_5H_{10}O_3$ (ISO-POOH/IEPOX), as they are not the focus of this study. A much lower concentration of $C_{10}$ accretion dimer products is present in the $NO_x$ cases (Fig. S4, Fig. S5) owing to the increased competition for $RO_2^\bullet$. A reduction in low volatility $C_{10}$ accretion products such as this has a large impact in the nucleation efficiency of this system when $NO_x$ is present[28]. The emergence of highly oxidised nitrogen-containing species in the two $NO_x$ cases is as expected; however, the ratio between the compounds containing two nitrogens is different between the NO case and the $NO_2$ case. The $NO_2$ case encompasses larger and more oxygenated compounds, most abundant dinitrogen containing compounds being $O_{9–10}$ and a weighted average N:O = 0.027 for all organics, compared to the most abundant oxygenated compounds being $O_8$ for the NO case, with a weighted average N:O = 0.05. This suggests the functional groups contain additional oxygen, consistent with a –$O_2NO_2$ peroxy nitrate functional group, rather than a nitrate –$ONO_2$ functional group. This difference is observable in the compounds containing one nitrogen as well. Focusing on first-generation products, we see similar behaviour shifting from the ISOPOOH-dominated environment to an ISOPOOH, IHN and IHPN split distribution and finally an ISOPOOH and IHPN environment, varying as expected from the mechanism outlined in Fig. 1. Given that product distributions vary with changes in the terminating radicals, the ratio of these radicals can be used to predict IP-OOM distribution without the need for an extensive chemical reaction scheme.

To highlight the efficiency of each termination pathway, we show the contribution of all second-generation compounds in Fig. 1 and $C_5H_{10}O_6$ to IP-OOM in Table 2. In combination with Fig. S5, it is clear that the presence of $NO_2^\bullet$ in the $NO_2$ case results in the formation of $RO_2NO_2$ whilst retaining the $IP_{0N}$ composition from the $HO_2$ case. In contrast, the NO case has a drastically different $IP_{0N}$ composition. With an increased $NO^\bullet$ concentration, the formation of $C_5H_{12}O_6$ is reduced, shifting the $IP_{0N}$ to a different distribution primarily dominated by $C_5H_{10}O_6$. Assuming $C_5H_{10}O_6$ is a hydroxy dihydroperoxy aldehyde, the volatility should be approximately an order of magnitude larger than $C_5H_{12}O_6$[27], contributing to a lesser extent in nucleation. This indicates that the presence of $NO^\bullet$ alters the underlying $RO_2^\bullet$ distribution, through the formation of alkoxy radicals via $RO_2^\bullet + NO^\bullet \rightarrow RO^\bullet + NO_2^\bullet$

**Table 2 | Contribution of second-generation products to $C_5$ IP-OOM total for each case**

| Compound | HO₂ case | fNO case | NO₂ case |
|---|---|---|---|
| $C_5H_{12}O_6$ | 18.8 | 0.1 | 7.7 |
| $C_5H_{10}O_6$ | 4.1 | 14.7 | 0.0 |
| $C_5H_{11}NO_7$ | – | 4.3 | 3.7 |
| $C_5H_{11}NO_8$ | – | 0.0 | 20.2 |
| $C_5H_{10}N_2O_8$ | – | 19.3 | 3.3 |
| $C_5H_{10}N_2O_9$ | – | 13.0 | 3.5 |
| $C_5H_{10}N_2O_{10}$ | – | 1.0 | 12.4 |

Calculated percentage contribution of $C_5H_{12}O_6$ (2 –OH, 2 –OOH), $C_5H_{10}O_6$ (–OH, 2 –OOH, C=O), $C_5H_{11}NO_7$ (2 –OH, –OOH, –ONO₂), $C_5H_{11}NO_8$ (2 –OH, –OOH, –O₂NO₂), $C_5H_{10}N_2O_8$ (2 –OH, 2 –ONO₂), $C_5H_{10}N_2O_9$ (2 –OH, –O₂NO₂, –ONO₂), and $C_5H_{10}N_2O_{10}$ (2 –OH, 2 –O₂NO₂) to $C_5$ IP-OOM for each case study. eg. Percentage of $C_5H_{12}O_6$ to $C_5$ IP-OOM = 100 × ([$C_5H_{12}O_6$]/[$C_5$ IP-OOM]).

unlike the $NO_2$ case, which preserves the underlying $RO_2^\bullet$ distribution through $RO_2^\bullet + NO_2^\bullet \rightleftharpoons RO_2NO_2$. Resultant $RO^\bullet$ formed from $NO^\bullet$ addition have a different fate than $RO_2^\bullet$, leading to a different $IP_{0N}$ distribution. Consequently, the $NO_2^\bullet$ pathway is a significant pathway to consider when determining the $NO_x$ impact on isoprene chemistry. However, further study would be required to understand the extent of the $NO^\bullet$ effect on the $RO_2^\bullet$ distribution. In addition, a different selection of nitrogen-containing species is present in the NO case compared to the $NO_2$ case, highlighting the transition from peroxynitrate to nitrate formation. Compounds described in Fig. 1 represent a plurality of IP-OOM in all three cases, and a deeper discussion on the remainder of the IP-OOM distribution is detailed in SI section D.

### The distribution of IP-OOM as a function of radical ratios
This study focuses on six termination products—$C_5H_{12}O_6$, $C_5H_{11}NO_7$, $C_5H_{10}N_2O_8$, $C_5H_{11}NO_8$, $C_5H_{10}N_2O_9$, $C_5H_{10}N_2O_{10}$—as described in Fig. 1. These compounds represent a substantial fraction of IP-OOM and play a pivotal role in new particle formation and early-stage particle growth under upper-tropospheric conditions[25,28]. Their formation involves complex multistep oxidation, where two sequential reactions of different $RO_2^\bullet$ species with $HO_2^\bullet$, $NO^\bullet$, or $NO_2^\bullet$ determine their final distribution.

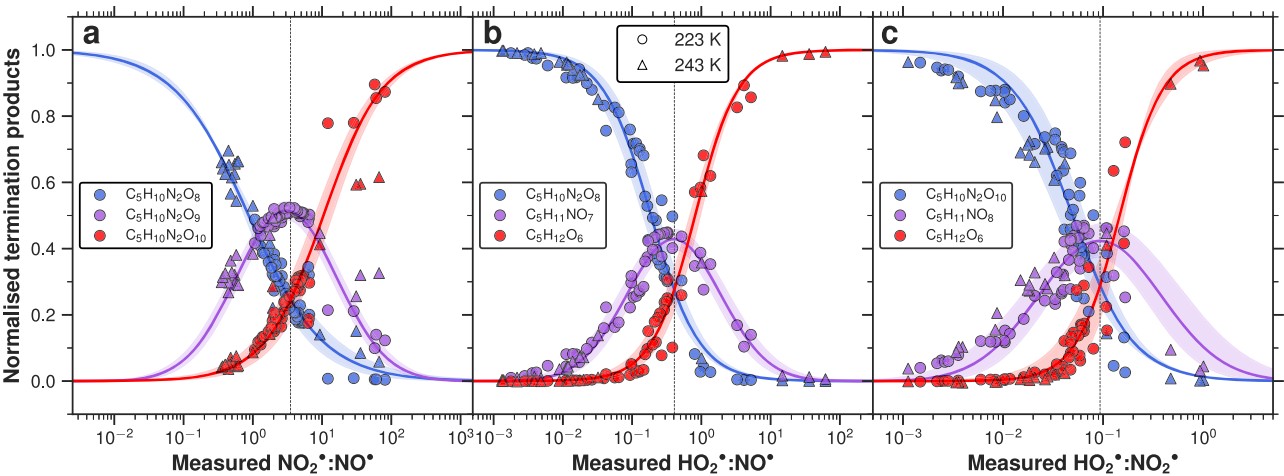

**Fig. 4 | Second-generation products as a function of radical ratios.** The normalised fraction (Eq. 5) of the three cross-branch products is plotted as a function of the respective branch radical ratios. For example, **a** shows the double NO• termination product $C_5H_{10}N_2O_8$ (blue), the mixed product $C_5H_{10}N_2O_9$ (purple), and the double $NO_2^•$ termination product $C_5H_{10}N_2O_{10}$ (red) normalised to the sum of all three, plotted against the $NO_2^•$:NO• ratio. **b** repeats the same treatment for $HO_2^•$:NO• and the second-generation products $C_5H_{10}N_2O_8$ (blue), $C_5H_{11}NO_7$ (purple) and $C_5H_{12}O_6$ (red), and **c** presents $C_5H_{10}N_2O_{10}$ (blue), $C_5H_{11}NO_8$ (purple) and

$C_5H_{12}O_6$ (red) as a function of $HO_2^•$:$NO_2^•$ ratios. Circles represent data at 223 K, and triangles at 243 K. Sigmoid and Gaussian curves have been fitted through the 223 K data and the standard deviation has been shaded, to represent uncertainty in the fit. Each data point in Fig. 4 is an averaged period of a steady-state chemistry stage achieved in CLOUD and vertical dashed lines indicate the point of equal termination for each branch, 3.51, 0.41 and 0.09 for (**a**, **b**, **c**), respectively. All second-generation concentrations are obtained from the NO$_3$-CIMS.

We have separated these six compounds into three branches, each containing three cross-branch products of a pair of radicals. The $NO_2^•$ and NO• cross-branch (Fig. 4a) includes the double NO• termination product ($C_5H_{10}N_2O_8$), the double $NO_2^•$ termination product ($C_5H_{10}N_2O_{10}$), and the mixed product, one NO• termination and one $NO_2^•$ termination, $C_5H_{10}N_2O_9$. The $HO_2^•$ and NO• cross-branch (Fig. 4b) describes the species of interest in Shen and Russell et al.[28] and Curtius et al.[25], $C_5H_{10}N_2O_8$ (dinitrate), $C_5H_{11}NO_7$ (mononitrate), and $C_5H_{12}O_6$ (dihydroperoxide). Finally, the $HO_2^•$ and $NO_2^•$ cross-branch (Fig. 4c) includes species with hydroperoxide and peroxynitrate functionality, $C_5H_{12}O_6$, $C_5H_{11}NO_8$, and $C_5H_{10}N_2O_{10}$. As shown in Table S2, the rate coefficients for these pathways differ by a factor of approximately 5–20, suggesting that variations in reactant concentrations can significantly influence the pathway branching ratios. To minimise uncertainties associated with speciated $RO_2^•$ and variations in oxidation rates, we normalise the three species as a fraction to the sum of all three (e.g. Eq. 5) for each cross-branch termination (NO•:$NO_2^•$, $HO_2^•$:NO• and $HO_2^•$:$NO_2^•$):

$$[C_5H_{12}O_6]_{norm} = \frac{[C_5H_{12}O_6]}{[C_5H_{12}O_6] + [C_5H_{11}NO_7] + [C_5H_{10}N_2O_8]} \quad (5)$$

Figure 4 illustrates that the relative fraction of the three cross-branch products varies with the ratio of the corresponding termination radicals at both 223 and 243 K. Our results suggest that the relative temperature dependence of radical termination with $HO_2^•$ and NO• between 223 and 243 K is minimal. The experiments conducted in the CLOUD chamber cover a broad range of radical ratios, with $NO_2^•$:NO• varying from 0.3 to 100, $HO_2^•$:NO• from 0.001 to 60, and $HO_2^•$:$NO_2^•$ from 0.001 to 1, as depicted in Fig. 4. Therefore, our data enables the investigation of the full chemical phase space in the cold upper troposphere.

Figure 4a depicts the changing ratio of the three di-nitrogen-containing species ($C_5H_{10}N_2O_8$, $C_5H_{10}N_2O_9$, and $C_5H_{10}N_2O_{10}$), described in Fig. 3. At high $NO_2^•$:NO•, $C_5H_{10}N_2O_{10}$ dominates the three di-nitrogen-containing species and gives way to the dinitrate $C_5H_{10}N_2O_8$

as $NO_2^•$:NO• is decreased, providing further evidence that $C_5H_{10}N_2O_{10}$ is a diperoxynitrate rather than a more oxidised dinitrate. Gaussian and sigmoid curves have been extrapolated to fit the 223 K data, and reach good agreement with the 243 K data. Interestingly, three 243 K data points at high $NO_2^•$:NO• fall off the $C_5H_{10}N_2O_{10}$ sigmoid fit of the 223 K data, lending to the other two. This could indicate that even at 243 K, there is thermal decomposition of $RO_2NO_2$; further study would be required to verify this.

Figure 4b spans the chemical phase space from a nitrate to hydroperoxide dominated environment and shows the ratio of $C_5H_{12}O_6$, $C_5H_{11}NO_7$ and $C_5H_{10}N_2O_8$—the result of double $HO_2^•$ termination, $HO_2^•$ and NO• termination, and double NO• termination, respectively—plotted against the radical ratio $HO_2^•$:NO•. As the $HO_2^•$:NO• ratio increases, the dominant pathway shifts from double NO• termination ($C_5H_{10}N_2O_8$, dinitrate) to the cross-product ($C_5H_{11}NO_7$, mononitrate), and ultimately to double $HO_2^•$ termination ($C_5H_{12}O_6$, dihydroperoxide). This trend reflects the competition between radicals under a limited supply of $RO_2^•$, underlining the critical role of radical ratios in governing the IP-OOM distribution. As shown in Fig. 4b, $C_5H_{11}NO_7$ only emerges as the dominant species within a narrow $HO_2^•$:NO• range of 0.2–0.6 (transition regime), whereas $C_5H_{10}N_2O_8$ dominates at lower ratios (<0.2) (nitrate regime) and $C_5H_{12}O_6$ prevails at higher ratios (>0.6) (hydroperoxide regime). In the transition regime, small changes in $HO_2^•$:NO• lead to quite drastic $C_5H_{12}O_6$, $C_5H_{11}NO_7$ and $C_5H_{10}N_2O_8$ changes and, therefore, IP-OOM composition changes.

The $HO_2^•$:$NO_2^•$ branch follows the same behaviour in Fig. 4c. Thus, providing more evidence this is peroxynitrate formation rather than a product of $NO_3^•$ oxidation as the latter would depend on both $HO_2^•$ and $NO_3^•$. The diperoxynitrate dominates the distribution at low $HO_2^•$:$NO_2^•$ and tends towards a dihydroperoxide-dominated environment at high ratios, confirming the reaction scheme shown in Fig. 1.

Consequently, the relative product distributions can be understood based on the measured $HO_2^•$, NO•, and $NO_2^•$ ratios. Conversely, given a known product distribution, the corresponding radicals can be inferred. By incorporating isoprene and OH• values to estimate $RO_2^•$ concentrations, it is possible to derive second-generation product

concentrations without relying on an extensive chemical scheme—an important advantage for global models.

It is important to note that the branching ratio ($\alpha$) of the reaction of $RO_2^\bullet$ with $NO^\bullet$, leading to organonitrate formation, appears to have a low-temperature, high-pressure asymptote[40]. We have used the parameterisation defined in Arey et al.[35] to determine this branching ratio. $\alpha_i$ depends on the number (i) of 'heavy atoms' in the $RO_2^\bullet$ species (non-hydrogen atoms present in the R-group of an $RO_2^\bullet$). Therefore, the intersection of curves obtained in Fig. 4 may differ for other $RO_2^\bullet$ with a different backbone.

For the analysis in Fig. 4, we used the isoprene oxidation product concentrations derived exclusively from the $NO_3$-CIMS measurements to ensure consistent instrument sensitivity and performance. As shown in Fig. S6, the Br-MION2-CIMS and the $NO_3$-CIMS exhibit excellent agreement for most detected species. Although the Br-MION2-CIMS demonstrates high sensitivity toward many species observed by the $NO_3$-CIMS, it did not detect all species of interest. A more detailed comparison of CIMS measurements is provided in SI section E.

We define the point of equal termination to represent the ratio of radical concentrations where the reactivities of $OH^\bullet$ with C=C followed by $RO_2^\bullet$ with $HO_2^\bullet$ and $NO^\bullet$ are equal. The corresponding termination point can be assumed to be approximately proportional to the ratio of effective rate coefficients of $HO_2^\bullet$ and $NO^\bullet$ with a $RO_2^\bullet$ under the given temperature and pressure conditions, $(k'_{HO2}[HO_2^\bullet])^2 \sim (k'_{NO}[NO^\bullet])^2$ and therefore, termination radical ratio = $[HO_2^\bullet]:[NO^\bullet] \sim k'_{NO}:k'_{HO_2}$. The effective rate coefficient here includes both the branching factor of the $RO_2^\bullet + NO^\bullet \rightarrow RONO_2$ pathway and the differing $NO^\bullet$ oxidation rates of first-generation intermediate products. Derived from the data in Fig. 4b, the experimental ratio is 0.41, an approximation for $k'_{NO} : k'_{HO_2}$. This value exceeds the literature range of 0.10–0.18 depending on $\alpha$, derived using rate coefficients from Wennberg et al.[5]. It is worth mentioning that the literature value is an extrapolation to low-temperature conditions, therefore, may not be accurate. However, we note our experiment is an approximation as it does not solve for the complicated steady-state kinetics, and further experiments are needed to constrain the ratio.

Completing similar analysis for the $NO_2^\bullet:NO^\bullet$ and $HO_2^\bullet:NO_2^\bullet$ cross-branches, gives equal termination points of 3.51 and 0.09, respectively.

$$\frac{k'_{NO}}{k'_{NO_2}} = 3.51, \quad \frac{k'_{NO}}{k'_{HO_2}} = 0.41, \quad \frac{k'_{NO_2}}{k'_{HO_2}} = 0.09 \quad (6)$$

$$k'_{NO_2} : k'_{HO_2} : k'_{NO} \sim 0.3 : 2.5 : 1 \quad (7)$$

Moreover, this analysis has been applied to first-generation products to highlight the importance of the radical ratios in determining IP-OOM distribution. The fraction of ISOPOOH and IHN as a function of $HO_2^\bullet:NO^\bullet$ is shown in Fig. S7. The ISOPOOH and IHN fraction remains relatively constant across most of the $HO_2^\bullet:NO^\bullet$ range until a critical threshold is reached, beyond which the fraction declines steeply. The transition in Fig. S7 occurs in the same radical ratio range as the transition regime in Fig. 4b, displaying the consistency of this result in both first- and second-generational product distribution. At equal concentrations of ISOPOOH and IHN, $HO_2^\bullet:NO^\bullet = 0.56$, which is in relatively good agreement with the termination radical ratio of second-generation species in Fig. 4b, 0.41. However, the critical $HO_2^\bullet:NO^\bullet$ decreases from 0.56 to 0.41 for first to second generation products. This could suggest the relative speed of the $HO_2^\bullet$ pathway relative to the $NO^\bullet$ pathway is faster for the second-generation than the first-generation oxidation. Steady-state kinetic derivations shown in SI

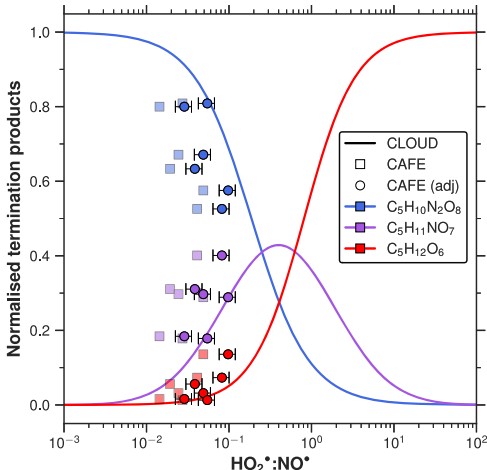

**Fig. 5 | Aircraft observation and laboratory comparison of the second-generation distribution as a function $HO_2^\bullet:NO^\bullet$.** Solid lines represent the experimental fits of $C_5H_{10}N_2O_8$ (blue), $C_5H_{11}NO_7$ (purple) and $C_5H_{12}O_6$ (red) from Fig. 4b of the CLOUD experiment, 223 K and 965 mbar. Square markers indicate atmospheric aircraft observations (CAFE) from Curtius et al.[25], specifically T4–T9, at 213 K and 187 mbar. To adjust for the pressure difference (see Fig. S8), circles (213 K and 950 mbar) represent CAFE T4–T9 periods, shifted by a factor of two in $HO_2^\bullet:NO^\bullet$ (CAFE (adj)). Error bars represent $1\sigma$ accuracy of the $HO_2^\bullet$ measurement[25].

section F indicate that this critical threshold is given by Eq. (8).

$$\left.\frac{HO_2}{NO}\right|^{TP} = \frac{k_{OH+ISOPOOH} \cdot k_{RO_2+NO} \cdot \alpha_6}{k_{OH+IHN} \cdot k_{RO_2+HO_2}} \quad (8)$$

Using reaction rate coefficients described in Table S2 from Wennberg et al.[5], the critical threshold is 0.13, whereas our experiment yields 0.56. Our results differ from the kinetic calculations expected for this radical ratio, hereby indicating an improvement could be made on the low-temperature extrapolation of $k_{OH+IHN}$, $k_{OH+ISOPOOH}$, $k_{RO_2+NO}$, and $k_{RO_2+HO_2}$.

## Laboratory and ambient comparison

To further compare our laboratory results with current understanding, ambient $HO_2^\bullet:NO^\bullet$ and second-generation compounds ($C_5H_{10}N_2O_8$, $C_5H_{11}NO_7$, $C_5H_{12}O_6$) for periods (T4–T9) from RF19 flight during the CAFE-Brazil campaign in Curtius et al.[25], have been incorporated into Fig. 5. These flight results cannot be directly compared due to a different temperature and pressure (215 K and 187 mbar). Therefore, we ran a kinetic model using the isoprene reaction mechanism from the supplementary information of Wiser et al.[41] (see 'Methods' for more detail) at different temperatures (223 and 213 K) and pressures (950 and 200 mbar), see Fig. S8, to study the effect these changes would have on the product distribution. The data in Fig. S8 qualitatively behaves similarly to the CLOUD data for all pressures and temperatures. Altering the temperature by 10 K leads to negligible shifts in the point of equal termination. However, changing the pressure from 950 to 200 mbar causes a large shift in the radical phase space and a shift in the point of equal termination from 0.15 to 0.07, roughly a factor of two, see Fig. S8. This adjustment is likely caused by the pressure dependent branching ratio of the $RO_2^\bullet + NO^\bullet$ reaction to form organonitrates. Therefore, the aircraft data were adjusted by a factor of two, to account for this pressure change, and fall in good agreement with our results. Additionally, it should be mentioned the $NO_3$-CIMS operated on the aircraft measured a combination of gas and particle phase due to an adiabatic heating in the inlet of the instrument as laid out in the SI of Curtius et al.[25]. Nevertheless, the notable

agreement between ambient measurements and our laboratory results indicates the chemical system achieved in CLOUD can accurately simulate the tropical upper troposphere.

Moreover, the aircraft radical ratio suggests that the upper-tropospheric conditions lie between the nitrate and transition regimes, with a predominance for $RO_2^\bullet + NO^\bullet$ over $RO_2^\bullet + HO_2^\bullet$, explaining the abundance of nitrates observed throughout this region[25].

The focus in Fig. 5 is on the $HO_2^\bullet$:$NO^\bullet$ branch due to the role of $C_5H_{12}O_6$, $C_5H_{11}NO_7$, and $C_5H_{10}N_2O_8$ in atmospheric particle formation[25,28] and the high photolysis of $NO_2^\bullet$ during the day. $NO_2^\bullet$:$NO^\bullet$ ratios from the same flight, range between 0.7 and 2.1 (see ED Table 1 Curtius et al.[25]), placing the system in a nitrate dominated regime—it is important to note that the $NO_2^\bullet$ measurement on the aircraft includes thermal artifacts from peroxynitrates. $C_5H_{12}O_6$, $C_5H_{11}NO_7$, and $C_5H_{10}N_2O_8$ all contribute to the growth of freshly nucleated particles due to their low volatility at 223 K as depicted in Fig. S9. $C_5H_{12}O_6$, $C_5H_{11}NO_7$ and $C_5H_{10}N_2O_8$ have calculated volatilities (using SIMPOL[7], see SI section G for more details) $6.5 \times 10^{-8}$, $1.8 \times 10^{-7}$, and $4.7 \times 10^{-7}$ $\mu g\,m^{-3}$, respectively. Volatilities measured with FIGAERO (see 'Methods') differ slightly from SIMPOL values: $1.4 \times 10^{-8}$ and $1.2 \times 10^{-6}$ $\mu g\,m^{-3}$ for $C_5H_{12}O_6$ and $C_5H_{11}NO_7$, respectively, highlighting the uncertainty of volatility calculations at 223 K. Importantly, these values are average volatilities for all sum formula isomers and variations can be up to a factor of 100 and 20 for structural and stereo isomers, respectively[42]. See SI section G for more details on their volatilities and oxidation states[43]. Moving from a low to high $HO_2^\bullet$:$NO^\bullet$ causes an increase in $C_5H_{12}O_6$ at the expense of $C_5H_{10}N_2O_8$. Similarly, Fig. S10 indicates moving from a low to high $HO_2^\bullet$:$NO^\bullet$ causes a significant increase in saturation ratio ($S_i^*$, see SI section G for details), thus supporting the more efficient nucleating capabilities observed in Shen and Russell et al.[28]. Deviation from this trend is observed when $NO_2^\bullet$:$NO^\bullet$ is greater than 10. At this ratio, the system lies in the diperoxynitrate regime of Fig. 4(a, c), $RO_2NO_2$ act as a reservoir of $RO_2^\bullet$ reducing the formation of $C_5H_{12}O_6$. Subsequently, the total saturation ratio of these species is decreased, which could have a large impact on the nucleation efficiency of the system. Therefore, highlighting the importance of peroxynitrate formation.

An $-ONO_2$ group lowers volatility similar to an $-OH$ group less than that for an $-OOH$ group[7,34], therefore analogous compounds, such as dihydroperoxide and dinitrate, have almost an order of magnitude difference in volatility. But both groups of compounds have low enough volatility to drive early growth at this temperature[28]. Contribution to atmospheric particle formation is a function of supersaturation, which is determined by the volatility and concentration of a compound, consequently, each decade higher volatility has to be compensated by approximately an order of magnitude increase in concentration to reach the same contribution to growth. Despite lower volatilities of non-nitrates, at low $HO_2^\bullet$:$NO^\bullet$ the saturation ratio is much greater than one (Fig. S10); therefore, in a supersaturated environment, these compounds would be driving atmospheric particle growth. Importantly, Shen and Russell et al.[28] showed that the driver of the formation and growth of new particles in the upper troposphere requires total $IP_{0N}$ and IP-OOM, respectively, including the $C_{10}$ accretion products. Therefore, it is likely a combination of higher volatility super-saturated nitrates with lower saturated, more efficiently nucleating non-nitrates are contributing, all enhanced by the presence of trace inorganic acids, and growth driven by the high concentration of low volatility species, regardless of nitrate functionality[25,28]. Ultimately, the fundamental mechanism underlying upper-tropospheric isoprene new particle formation entails the consecutive addition of two $OH^\bullet$ radicals to the isoprene backbone, without the occurrence of fragmentation, thereby forming IP-OOM, irrespective of the termination radicals.

## Global implication

In order to ascertain the importance of the radical branching of IP-OOM formation in the atmosphere, we have used the ECHAM/MESSy Atmospheric Chemistry (EMAC) model[44] (see 'Methods' for more details) for an annual global simulation from March 2022 to February 2023. EMAC concentrations of isoprene and radicals have been compared above the Amazon rainforest and evaluated against aircraft measurements from the CAFE-Brazil campaign[45], confirming in the model a diurnal cycle of isoprene in the boundary layer and the largest isoprene mixing ratios in the upper troposphere found during the early morning. We will be considering an annual average, therefore, it is important to remember that seasonality has been removed, therefore different radical ratios may be observed in specific seasons. For example, the wet season in the upper troposphere over the Amazon would expect more frequent deep convection, resulting in higher levels of isoprene and subsequent $NO_x$ transported than the dry season, see Fig. S11.

Although upper-tropospheric isoprene + $OH^\bullet$ oxidation occurs across the globe (Fig. 6a), there are three hotspots for this reaction corresponding to the three largest tropical rainforests (1) Amazon, (2) Congo and (3) Indonesia, Papua New Guinea and Northern Australia. Tropical rainforests frequently form deep convective systems due to a combination of high surface temperatures and humidity, efficiently transporting isoprene from the forest up to upper-tropospheric altitudes.

For the Amazon rainforest, we have created three different daytime scenarios: (b, e) sunrise, (c, f) midday and (d, g) sunset to investigate the daily evolution of IP-OOM chemistry using the isoprene oxidation rate and the $HO_2^\bullet$:$NO^\bullet$ ratio. Nocturnal accumulation of isoprene in the upper troposphere (Fig. S12) due to slow oxidation by $NO_3^\bullet$ and $O_3$ leads to a large area in the morning where sufficient isoprene oxidation can occur. A combination of fast daytime isoprene $OH^\bullet$-oxidation in the boundary layer and horizontal transport replacing airmasses with high isoprene concentrations with airmasses from marine or non-forested regions that have low isoprene content lead to this area decreasing throughout the day, as seen across Fig. 6(b, c, d) and in the isoprene concentrations for each scenario in Fig. S13. Therefore, the total rate of isoprene oxidation (total IP-OOM production rate) above the Amazon increases dramatically at sunrise (Fig. S12a), reaching a maximum around 10:00 local time, exhibiting a diurnal cycle and in agreement with observations[25]. There are lower isoprene concentrations at midday (Fig. S12b and Fig. S13); however, an increased $OH^\bullet$ concentration gives rise to larger localised IP-OOM production rates as shown across Fig. 6(b, c, d). Consequently, IP-OOM formation should occur throughout the day, however, due to a lower condensation sink this will lead to primarily nucleation in the morning and condensation on pre-existing aerosol in the afternoon.

A similar diurnal cycle of $HO_2^\bullet$:$NO^\bullet$ is observed in Fig. 6(e, f, g) and Fig. S12a. $HO_2^\bullet$ levels follow the increased levels of $OH^\bullet$ due to increased solar flux, whereas the build of $NO_x$ overnight gets depleted throughout the day resulting in a decreasing $NO^\bullet$ concentration. Changes of $HO_2^\bullet$:$NO^\bullet$ throughout the day indicates a shift from the nitrate regime (nitrate-dominated termination) towards the transition regime, a mixed nitrate-hydroperoxide situation at midday. The $HO_2^\bullet$:$NO^\bullet$ observed in the model throughout the day ranges from 0.06 to 0.5, therefore always residing in a $NO^\bullet$-dominated termination region but with an increased contribution from the mononitrate cross-product at midday when the $HO_2^\bullet$:$NO^\bullet$ reaches a maximum. The relative predicted fluxes clearly depict this in Fig. S14. The total product formation increases at midday as explained in Fig. S12. Simultaneously, the relative distribution changes from $C_5H_{10}N_2O_8$-dominated to $C_5H_{11}NO_7$-dominated. This can help explain the dominance of nitrates observed in Curtius et al.[25]. This behaviour is consistent for all three tropical regions, evident in Fig. S15.

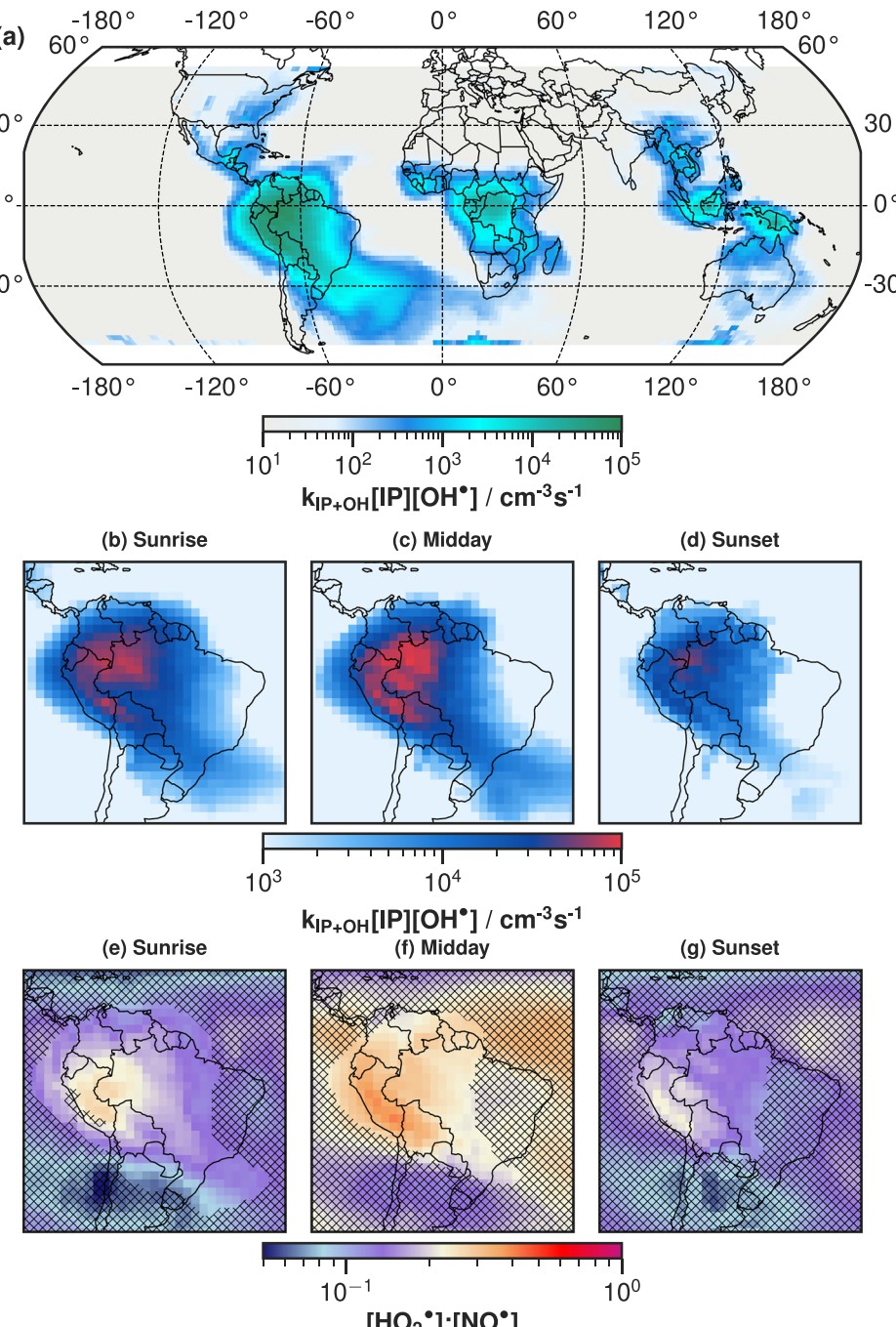

**Fig. 6 | Global simulation of upper-tropospheric IP-OOM chemistry. a** The global daytime reactivity of isoprene + OH$^{\bullet}$. Using annually averaged daytime isoprene and OH$^{\bullet}$ concentrations from the EMAC global model and a reaction rate coefficient $k_{IP} = 2.7 \times 10^{-11}\, e^{\frac{390}{T}}\, cm^3\, s^{-1}$[5]. Focusing on the Amazon rainforest, isothermal-surface plots depict the reactivity of isoprene with OH$^{\bullet}$ at three daytime periods **b** Sunrise, **c** Midday and **d** Sunset. The HO$_2^{\bullet}$:NO$^{\bullet}$ ratio at three daytime periods **e** Sunrise, **f** Midday and **g** Sunset is also plotted for each daytime period. A hashed grid (masking insufficient oxidation) has been added so to highlight regions with isoprene and OH$^{\bullet}$ levels high enough that sufficient oxidation could occur: isoprene $>2.4 \times 10^7\, cm^{-3}$ (223 K, 230 mbar) and OH$^{\bullet}$ $>6 \times 10^5\, cm^{-3}$ (223 K, 230 mbar). Midday is defined using the hour of highest shortwave flux at the top of the atmosphere, whereas sunrise and sunset are defined when the shortwave flux is closest to a quarter of the maximum shortwave flux. All data are annually averaged isothermal-surface plots at the temperature of 223 K from the global model (EMAC), ranging between 200 and 300 hPa and (−51.3°, 51.3°) latitude. A detailed description of the model conditions is presented in 'Methods'. The map in the figures were made with Natural Earth.

Rainforests act as large sources of isoprene, which deep convective systems can transport almost constantly up to the upper troposphere, where OH$^{\bullet}$ formation is efficient, highlighting these as important regions for the low-temperature IP-OOM oxidation scheme. Moreover, the low temperatures explored in this study occur throughout the upper troposphere, see Fig. S16, and reach as low as 5 km at higher latitudes. We have confirmed that the chemical system and phenomenon described in refs. 25, 28 occurs globally very frequently, and will act as a source of particles throughout the upper troposphere[29]. In addition, we expect that the other tropical rainforests would exhibit similar isoprene oxidation and thus, new particle formation.

There is an innate connection between deep convection and lightning; therefore, the transported isoprene will be accompanied by lightning-generated $NO_x$, leading to $OH^\bullet$ recycling from the reaction of $HO_2^\bullet + NO^{\bullet}$[46] and a predominance of nitrates observed in these regions. This underlines the importance of the $NO^\bullet$ termination pathway in the tropical upper troposphere. We have shown the increased contribution of the $HO_2^\bullet$ termination pathway at midday under upper-tropospheric conditions. Furthermore, at lower outflow regions, there may be less $NO_x$ produced as the parcel of air will have spent less time in a lightning environment. In this case, a higher $HO_2^\bullet:NO^\bullet$ would push the system towards the hydroperoxide regime, increasing the saturation ratio. Therefore, despite outflow occurring at warmer temperatures, IP-OOM oxidation may still drive new particle formation and growth.

Under daytime upper-tropospheric conditions, photolysis of $NO_2^\bullet$ to $NO^\bullet$ is rapid, therefore, the annually averaged $NO_2^\bullet:NO^\bullet$ ranges between 0.1 and 0.4 in the tropics and up to ~1 at mid-latitudes (see Fig. S17). However, this ratio would be subject to a high seasonal variability; therefore, values at higher latitudes could be much larger or smaller, dependent on the time of the year. This implies that $RO_2NO_2$ formation will occur under these conditions, although low $NO_2^\bullet:NO^\bullet$ would result in a nitrate-dominated regime. As observed in Fig. 3 of Curtius et al.[25], $C_5H_{10}N_2O_8$ (dinitrate) dominates the $IP_{2N}$ with a small contribution from the mixed-product $C_5H_{10}N_2O_9$ (nitrate and peroxynitrate). Therefore, confirming the importance of $RO_2NO_2$ in the upper troposphere. Furthermore, our analysis in Fig. 4 has eliminated the speciated $RO_2^\bullet$, therefore, similar analysis can be applied to other isoprene systems. For instance, isoprene $RO_2^\bullet$ formed through $NO_3^\bullet$ addition during the night would be subject to large concentrations of $NO_2^\bullet$ forming $RO_2NO_2$—this may become important when considering the transport of organic species across large distances. $RO_2NO_2$ formed throughout the night can sequester isoprene $RO_2^\bullet$ radicals, preventing further termination and transport mass until it descends, warms and releases the $RO_2^\bullet$, similar to PAN. Mass transportation of isoprene $RO_2^\bullet$ like this could significantly affect the chemical distribution in a region far removed from organic emissions. For example, depositing $RO_2^\bullet$ into a region of high inorganic acid could lead to significant growth as transported organics can provide the mass for nucleating inorganic systems.

Other organic species, such as monoterpenes, have been observed in upper-tropospheric outflow[25,26] at much lower concentrations. The presence of monoterpenes would increase $RO_2^\bullet$ competition and form analogous ROOH, $RONO_2$ and $RO_2NO_2$, in addition to $C_{15}$ accretion products. Increased nighttime reactivity of monoterpenes with $NO_3^\bullet$ could lead to an environment favourable for $RO_2NO_2$ formation. Additionally, due to the abundance of isoprene and the fast reactions with $OH^\bullet$, it is expected that the isoprene chemical system observed in these experiments will be relatively unaffected, however, elevated monoterpene $RO_2^\bullet$ concentrations would alter atmospheric radical ratios due to differing reactivities with $NO^\bullet$, $HO_2^\bullet$ and $NO_2^\bullet$ compared to isoprene $RO_2^\bullet$. A follow-up study on the impact of monoterpenes on new particle formation and chemical distribution under upper-tropospheric conditions is warranted to confirm these predictions.

Overall, the rapid $OH^\bullet$-initiated oxidation of isoprene in the upper troposphere plays a crucial role in new particle formation, significantly contributing to the formation of cloud condensation nuclei in this region. Our experiments, conducted at 243 and 223 K, reveal that at such low temperatures, the formation of $RO_2NO_2$ becomes a significant pathway in determining the fate of isoprene $RO_2^\bullet$, alongside reactions with $NO^\bullet$ and $HO_2^\bullet$. Thermal decomposition is greatly suppressed at low temperatures, and thus, we were able to estimate a reaction rate coefficient of $RO_2^\bullet + NO_2^\bullet$, $k_{NO_2} = 3.4 \times 10^{-12}\ cm^3\ s^{-1}$. $RO_2NO_2$ serves as a reservoir of $RO_2^\bullet$ and could be transported to lower altitudes, where it decomposes, reshaping $RO_2^\bullet$ distribution and

influencing aerosol formation. We found that two rapid $OH^\bullet$ oxidations to the isoprene backbone drive IP-OOM formation. We observed an IP-OOM yield of 68% from the reacted precursors (ISOPOOH, IEPOX, IHN and IHPN) and $OH^\bullet$ highlighting the speed of the IP-OOM precursor oxidation.

In Fig. 1, we propose a simplified isoprene oxidation mechanism for upper-tropospheric conditions, emphasising the importance of radical termination on the IP-OOM distribution. By varying radical concentrations, we found a direct connection between the radical ratios and second-generation oxidation product distributions. For example, the ratio of $HO_2^\bullet$ to $NO^\bullet$ can be used to estimate the distribution of $C_5H_{12}O_6$, $C_5H_{11}NO_7$, and $C_5H_{10}N_2O_8$—important compounds for upper-tropospheric new particle formation[25,28]. Since the relative abundances of condensing species are governed by the reactivities of the corresponding reaction pathways. Low $HO_2^\bullet:NO^\bullet$ yields a nitrate-dominated regime, whereas a high $HO_2^\bullet:NO^\bullet$ creates a hydroperoxide-dominated regime. A transition regime between these two exists where reactivities are similar and small changes in $HO_2^\bullet:NO^\bullet$ can yield drastic IP-OOM composition changes. Although this study focuses on six specific oxidation products, moving from low to high $HO_2^\bullet:NO^\bullet$ indicates a shift from $RO_2^\bullet + NO^\bullet$ reactivity towards $RO_2^\bullet + HO_2^\bullet$ for all $RO_2^\bullet$.

Using radical ratios when product distributions and radical reactivities are equal, our results provide an estimate for the ratio of effective reaction rate coefficients, $k'_{NO_2} : k'_{HO_2} : k'_{NO}$ to be 0.3:2.5:1 at 223 K, providing radical reactivity ratios under upper-tropospheric conditions and highlighting an improvement could be made to current kinetic extrapolations.

In addition, we obtained a good agreement with aircraft observations[25], confirming our results as valid simulations of upper-tropospheric conditions.

Under daytime upper-tropospheric conditions, $NO^\bullet$ and $HO_2^\bullet$ are the most important terminators and create a primarily nitrate-dominated environment, owing to the excess of $NO^\bullet$ from lightning. However, the $HO_2^\bullet$ pathway becomes increasingly more important as the sun reaches its maximum. Under nighttime conditions, we expect the formation of $RO_2NO_2$ at these low temperatures will become a significant pathway in regions with larger $NO_2^\bullet:NO^\bullet$ ratios, and act as a reservoir for $RO_2^\bullet$.

Global model results indicate daytime isoprene oxidation occurs throughout the tropical upper troposphere, creating a large source of IP-OOM during the day. Isoprene nitrates will be more abundant under upper-tropospheric conditions; however, non-nitrate compounds have a lower volatility than their nitrate counterparts. Hence, the crucial factor defining new particle formation and growth in the tropical upper troposphere is the formation of second-generation isoprene oxidation products, from two $OH^\bullet$ oxidations regardless of the termination radical[25,28].

## Methods
### CLOUD chamber
All experiments were carried out in the 26.1 $m^3$ stainless steel CERN CLOUD chamber[39]. A thermal housing surrounding the chamber maintained the temperature at 223 and 243 K with a stability of around 0.1 K[47]. The pressure inside the chamber was kept at 5 mbar above atmospheric pressure (965 mbar mean pressure). Synthetic air, 79% $N_2$ and 21% $O_2$, combined with trace gases was injected into the chamber at ~340 litres per minute to compensate for sampling losses to the analysing instruments attached to the chamber.

Isoprene nucleation experiments involved continuous injection of isoprene and generation of $OH^\bullet$ radicals from the photolysis of $O_3$ using a combination of four 200 W Hamamatsu Hg-Xe lamps and a 52 W low-pressure mercury lamp centred on 254 nm non-$NO_x$ experiments. For $NO_x$ experiments, $H_2O_2$ or HONO photolysis generated $OH^\bullet$ radicals, using a combination of the low-pressure mercury lamp and a

UV light source centred on 385 nm. To achieve different $NO_x$ concentrations, $NO_2^\bullet$ was injected and photolysed by the UV light source centred on 385 nm. To remove contaminants, such as monoterpenes, the isoprene vapour was passed through a cryo-trap, where low volatility contaminants were frozen out. $SO_2$ and $H_2SO_4$ were also present during the experiments to study the impact of inorganic acid on nucleation.

Typical experiments to measure nucleation and growth rates lasted several hours with fixed experimental conditions, during which steady-state chemistry was achieved. The experimental values reported here correspond to the average values measured during steady-state conditions.

### Measurements of IP-OOM

This paper uses data from three mass spectrometers ($NH_4$-PTR3-CIMS[48], Br-MION2-CIMS[49], $NO_3$-CIMS) with different ionisation techniques ($NH_4^+$, $Br^-$ and $NO_3^-$) to measure the oxidation products of isoprene. Each chemical ionisation technique has a different sensitivity to analytes; the species contribution from each instrument to the total sum of IP-OOM follows the genetic algorithm described in Shen and Russell et al.[28] and outlined as follows. Each instrument characterised peaks observed distinct from the background and created time series for their concentrations. If an IP-OOM is detected by only one instrument, it is added to the sum directly. If two or more instruments detect the same sum formula, the same species is cross-compared between instruments. If the same species yields a correlation coefficient greater than 0.5, they are considered to be the same molecule and the instrument with the largest signal is added to the sum—since assuming an $H_2SO_4$ calibration factor for every peak in the $NO_3$-CIMS and the Br-MION2-CIMS can lead to underestimating species with a lower detection efficiency. If the species yields a correlation coefficient lower than 0.5, then they are determined to be distinct isomers and both are added to the IP-OOM sum.

The $NH_4$–PTR3-CIMS was calibrated using Hexanal under different temperatures and relative humidity[50], unless a gas standard was available. In order for the $NO_3$-CIMS to be able to confidently measure nitrate species distinct from its reagent ion, the instrument was run with labelled nitric acid ($H^{15}NO_3$). Both the $NO_3$-CIMS and Br-MION2-CIMS were calibrated for $H_2SO_4$[51], giving calibration factors of $1 \times 10^{10}$ cm$^{-3}$ and $1.9 \times 10^{10}$ cm$^{-3}$ for $NO_3$-CIMS and Br-MION2-CIMS, respectively, with a systematic error in the measurement of $H_2SO_4$ ranging from 88 to 120%[28].

### Termination radical measurements

$HO_x$ radical concentrations were measured with a HydrOxyl Radical measurement Unit based on fluorescence Spectroscopy (HORUS)[52], the systematic error of the measurement for $OH^\bullet$ and $HO_2^\bullet$ is 12 and 30%, respectively. $HO_2^\bullet$ measurements were also made with the Br-MION2-CIMS through the $HO_2^*Br^-$ peak and a cross-calibration between these two measurements was performed[28].

$O_3$ was measured using a Thermo Environmental Instruments, TEI 49C gas monitor.

$NO_x$ radicals, $NO^\bullet$ and $NO_2^\bullet$ concentrations were measured by two separate instruments: $NO^\bullet$ with a chemiluminescence detector sensitive to the emission from $NO^\bullet + O_3$ (ECO PHYSICS, CLD 780TR). The instrument was calibrated by another $NO^\bullet$ monitor (ECO PHYSICS, CLD 780TR), which had been calibrated using a CMK5 Touch dilution system (Umwelttechnik MCZ GmbH) with an $NO^\bullet$ bottle (Praxair, 1 ppmv in $N_2$) and synthetic air. Background values of the $NO^\bullet$ monitor used in the study have been subtracted. $NO_2^\bullet$ was measured using a Cavity Attenuated Phase Shift nitrogen dioxide monitor (CAPS $NO_2$, Aerodyne Research Inc.). The instrument undergoes a 5-min background measurement every hour. The finalised $NO_2^\bullet$ time series has been background corrected and an interpolation has been applied to give a continuous time series during background measurements.

### FIGAERO

A Filter Inlet for Gas and AEROsol (FIGAERO)[53] CIMS coupled to $I^-$ chemical ionisation was used to determine the particle phase composition of IP-OOMs and can give approximate volatility measurements based on the temperature of desorption.

### Kinetic model

Using a combination of ozone and $NO_x$ can generate nitrate radicals ($NO_3^\bullet$), at CLOUD, there is no measurement of $NO_3^\bullet$, therefore, we used a kinetic model to estimate $NO_3^\bullet$ concentrations using known precursor concentrations of $O_3$, $NO^\bullet$, $NO_2^\bullet$, $OH^\bullet$, $HO_2^\bullet$ and isoprene. Additionally, the kinetic model was used to generate an understanding of the effect of pressure and temperature on second-generation oxidation products.

The reaction mechanism used in this study combines the isoprene reaction mechanism from the supplementary of Wiser et al.[41] and supplementary $NO_x$ and $O_x$ reactions; the full reaction file can be found in the supplementary files. The total reaction mechanism consists of 1359 reactions, and was run with temperatures set to 223 K, relative humidity at 70% and pressure set to 950 mbar.

For predicting $NO_3^\bullet$ concentrations, the model was run with isoprene and carbon monoxide concentrations fixed at 1 and 100 ppb, respectively. Concentrations of $OH^\bullet$, $O_3$, $NO^\bullet$, $NO_2^\bullet$ and $HO_2^\bullet$ were fixed at their measured values for each stage. The model was used to solve the complex kinetic equations and the instantaneous concentration $NO_3^\bullet$ was taken as an approximation to the real value. The kinetic model does not include any physical losses that are inherent to CLOUD. Photolysis and light intensity have been estimated to replicate CLOUD; their values are included in the reaction file. The $NO_3^\bullet$ concentrations should not be used quantitatively but rather as an indication of high $NO_3^\bullet$ periods.

To provide a pressure and temperature correction, the model was run for 1000 s with $OH^\bullet$ fixed at 0.5 ppt, isoprene and carbon monoxide concentrations initially set at 1 and 100 ppb, respectively. Additionally, $HO_2^\bullet$ and $NO^\bullet$ were fixed at different concentrations for each experiment and $NO^\bullet$ was ramped in order to span a wide range of $HO_2^\bullet:NO^\bullet$ ratios. This analysis was replicated two more times with 213 K and 950 mbar, and 223 K and 200 mbar to create Fig. S8, exploring the effect of temperature and pressure on the system. Within the isoprene reaction mechanism, there exist many structural isomers of $C_5H_{12}O_6$, $C_5H_{11}NO_7$ and $C_5H_{10}N_2O_8$, therefore, they were all summed based on sum formula, as would be seen in a mass spectrometer. The number of isomers and names for each species in the kinetic model is listed in Table S4.

### Global model

We employed the ECHAM/MESSy Atmospheric Chemistry (EMAC) model[44] for global simulations of upper-tropospheric composition. EMAC consists of the base general circulation model ECHAM v5.3.02[54] coupled to various physics, chemistry and diagnostic routines, modularised in the second generation of the Modular Earth Submodel System (MESSy)[55]. Simulations are performed at T63 horizontal resolution (~190 × 190 km at the equator) and with 90 vertical levels up to 0.01 hPa, with the model's dynamics weakly 'nudged' towards meteorological reanalysis data (ERA5)[56].

Gas phase chemistry is treated with the MECCA submodel[57], using MIM1 chemistry[58], including approximately 120 gas phase species and 300 chemical reactions—the proposed chemistry described in this study has not been included in the simulation. This study uses the simulated atmospheric radical ratios to provide a global implication of the chemistry explored here; a follow-up study is planned on modelling the full implementation of this low-temperature chemistry. Isoprene mixing ratios over the Amazon basin have been evaluated with measurements from the CAFE-Brazil campaign[45]. Anthropogenic emissions are sourced from the CEDS

database[59], while biogenic emissions are calculated online using MEGAN[1]. $NO_x$ emissions from lightning are estimated with the algorithm developed by Grewe et al.[60], with total emission of ~6.2 Tg N yr$^{-1}$, demonstrating best agreement with observations and falling within the estimate of 2–8 Tg N yr$^{-1}$[61].

This study simulated the year 2022 (1st March 2022–28th February 2023)[45], with hourly model output between −51.3° and 51.3° latitude, sampled along isothermal surfaces of 223 K (−50 °C). The presented data show annually averaged atmospheric conditions at midday (maximum short-wave flux at the top of the atmosphere), and at sunrise and sunset (25% of maximum short-wave flux at the top of the atmosphere, respectively).

## Data availability
Data for all figures in the main text and supplementary materials are publicly available at the Zenodo repository (https://doi.org/10.5281/zenodo.16276710)[62].

## Code availability
The Modular Earth Submodel System (MESSy, https://doi.org/10.5281/zenodo.8360186)[63] is continuously further developed and applied by a consortium of institutions. The usage of MESSy and access to the source code is licensed to all affiliates of institutions which are members of the MESSy Consortium. Institutions can become a member of the MESSy Consortium by signing the MESSy Memorandum of Understanding. More information can be found on the MESSy Consortium Website (http://www.messy-interface.org).

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

## Acknowledgements

We thank CERN for supporting CLOUD with important technical and financial resources and for providing a particle beam from the CERN Proton Synchrotron. Funding: This work was supported by the European Union's MSCA Doctoral Network CLOUD-DOC 101073026 (D.M.R., A.C., A.M., B.Y., C.X., G.R.U., H.B., I.Z., M.K.S., P.R., W.Y. and Z.Z.). Federal Ministry of Education and Research (BMBF) financed project CLOUD-22 01LK2201A (L.C.-P., J.C.), 01LK2201B (O.M.), 01LK2201C (G.R.U.). Research Council of Finland grant numbers 346371 (S.S.) and 359331 (X.-C.H.). US National Science Foundation awards AGS-2215522 (R.V.), AGS-2027252 (R.V.), AGS-2215527 (R.C.F.), AGS-2431817 (N.M.D.), CHE-2336463 (J.De., N.M.D.), AGS-2215489 (N.M.D.). Swiss National Science Foundation grants 200021_213071 (I.E.H.) and 216181 (L.D.). European Union's Horizon 2020 research and innovation programme under grant agreement No. 856612 (T.C.). Estonian Research Council projects PRG2738 (H.J.), RVTT3 (H.J.), TT18 (H.J.).

## Author contributions

D.M.R., J.S., X.-C.H., J.K. and J.C. planned the experiments. D.M.R., F.K., J.S., J.De., N.B., H.K., L.C.-P., M.S., A.A., A.C., A.M., A.O., A.T., B.J., B.M., B.Y., E.A., E.S., H.B., I.Z., J.A., J.T., J.W., J.Z., L.D., L.L., M.B., M.K.S., M.M., M.R., M.Z.-W., N.S.U., P.R., P.K., R.C.-S., R.X.W., T.K., T.P., W.S., W.Y., Y.T., Z.Z., A.H., D.R.W., H.J., M.Ku., K.L., P.M.W., R.C.F, R.V., S.S., T.C., N.M.D., H.H., J.K., X.-C.H. and J.C. prepared the CLOUD facility or measuring instruments. D.M.R., F.K., J.S., M.Ko., J.De., N.B., C.X., H.K., L.C.-P., M.S., A.A., A.C., A.T., B.J., B.M., B.Y., E.S., G.R.U., H.B., J.A., J.T., J.W., J.Z., L.D., L.L., M.R.C., M.R., M.Z.-W., N.S.U., P.H., P.R., R.C.-S., R.X.W., S.R., T.K., W.S., W.Y., Y.T., Z.Z., H.H., J.K. and X.-C.H. collected the data. D.M.R., F.K., J.S., M.Ko., J.De., C.X., L.C.-P., B.M., B.Y., J.W., L.L., M.R.C., P.H., W.S., Y.T., Z.Z. and X.-C.H. analysed the data. M.Ko., C.X., T.C. and A.P. carried out the global model simulations. D.M.R. wrote the manuscript with contributions from J.S., X.-C.H., N.M.D., M.Ko., J.K. and J.C.. Finally, D.M.R., F.K., J.S., M.Ko., J.De., N.B., C.X., H.K., M.S., S.R., I.E.H., J.L., O.M., R.V., S.S., A.P., N.M.D., J.K., X.-C.H. and J.C. commented on and edited the manuscript.

## Funding

## Competing interests

The authors declare no competing interests.

## Additional information

Douglas M. Russell [1] ✉, Felix Kunkler [2], Jiali Shen [3,4], Matthias Kohl [2], Jenna DeVivo [5,6], Nirvan Bhattacharyya [5,6], Christos Xenofontos [7], Hannah Klebach [1], Lucía Caudillo-Plath[1], Mario Simon[1], Emelda Ahongshangbam [3,8], João Almeida[9,10], Antonio Amorim [10,11], Hannah Beckmann [12], Mattia Busato[9], Manjula R. Canagaratna[13], Anouck Chassaing [14], Romulo Cruz-Simbron[15,16], Lubna Dada [17], Philip Holzbeck [2], Bernhard Judmaier[18], Milin Kaniyodical Sebastian[19], Paap Koemets [12], Timm Krüger[1], Lu Liu [17], Monica Martinez [2], Bernhard Mentler [18], Aleksandra Morawiec[20], Antti Onnela[9], Tuukka Petäjä [3], Pedro Rato[1,9], Mago Reza[15,16], Samuel Ruhl[2], Wiebke Scholz[18], Eva Sommer [9,20], António Tomé [21], Yandong Tong[15,16], Jens Top [17], Nsikanabasi Silas Umo[19,22], Gabriela R. Unfer [23], Ryan X. Ward [24], Jakob Weissbacher[18], Boxing Yang [17], Wenjuan Yu [3], Marcel Zauner-Wieczorek [1], Imad Zgheib [25], Jiangyi Zhang [3], Zhensen Zheng[18,26], Imad El Haddad [17], Richard C. Flagan [24], Armin Hansel [18], Heikki Junninen [12], Markku Kulmala [3,4,27,28], Katrianne Lehtipalo [3,29], Jos Lelieveld [2,7], Ottmar Möhler [19], Siegfried Schobesberger [30], Rainer Volkamer [15,16], Paul M. Winkler [20], Douglas R. Worsnop [3,13], Theodoros Christoudias [7], Andrea Pozzer [2,7], Neil M. Donahue [5,6,31], Hartwig Harder [2], Jasper Kirkby [1,9], Xu-Cheng He [3,32] & Joachim Curtius [1] ✉

[1]Institute for Atmospheric and Environmental Sciences, Goethe University Frankfurt, Frankfurt am Main, Germany. [2]Atmospheric Chemistry Department, Max Planck Institute for Chemistry, Mainz, Germany. [3]Institute for Atmospheric and Earth System Research/Physics, Faculty of Science, University of Helsinki, Helsinki, Finland. [4]Helsinki Institute of Physics, University of Helsinki, Helsinki, Finland. [5]Department of Chemistry, Carnegie Mellon University, Pittsburgh, PA, USA. [6]Center for Atmospheric Particle Studies, Carnegie Mellon University, Pittsburgh, PA, USA. [7]Climate and Atmosphere Research Center (CARE-C), The Cyprus Institute, Nicosia, Cyprus. [8]Department of Chemistry, University of Helsinki, Helsinki, Finland. [9]CERN, European Organisation for Nuclear Research, Geneva, Switzerland. [10]Faculdade de Ciências da Universidade de Lisboa, Lisboa, Portugal. [11]Laboratório de Instrumentação e física experimental de Partículas, Lisboa, Portugal. [12]Institute of Physics, University of Tartu, Tartu, Estonia. [13]Aerodyne Research Inc., Billerica, MA, USA. [14]Department of Environmental Science, Stockholm University, Stockholm, Sweden. [15]Department of Chemistry, University of Colorado Boulder, Boulder, CO, USA. [16]Cooperative Institute for Research in Environmental Sciences, University of Colorado Boulder, Boulder, CO, USA. [17]PSI Center for Energy and Environmental Sciences, Paul Scherrer Institute, Villigen, Switzerland. [18]Institute for Ion Physics and Applied Physics, University of Innsbruck, Innsbruck, Austria. [19]Institute of Meteorology and Climate Research, Atmospheric Aerosol Research, Karlsruhe Institute of Technology, Karlsruhe, Germany. [20]Faculty of Physics, University of Vienna, Wien, Austria. [21]Instituto Dom Luiz (IDL), Universidade da Beira Interior, Covilhã, Portugal. [22]Department of Chemistry and Biochemistry, University of North Carolina Wilmington, Wilmington, North Carolina, USA. [23]Atmospheric Microphysics Department, Leibniz Institute for Tropospheric Research (TROPOS), Leipzig, Germany. [24]Division of Chemistry and Chemical Engineering, California Institute of Technology, Pasadena, CA, USA. [25]TOFWERK, Thun, Switzerland. [26]IONICON Analytik GmbH, Innsbruck, Austria. [27]Joint International Research Laboratory of Atmospheric and Earth System Sciences, School of Atmospheric Sciences, Nanjing University, Nanjing, China. [28]Aerosol and Haze Laboratory, Beijing Advanced Innovation Center for Soft Matter Science and Engineering, Beijing University of Chemical Technology, Beijing, China. [29]Finnish Meteorological Institute, Helsinki, Finland. [30]Department of Technical Physics, University of Eastern Finland, Kuopio, Finland. [31]Department of Chemical Engineering, Carnegie Mellon University, Pittsburgh, PA, USA. [32]Yusuf Hamied Department of Chemistry, University of Cambridge, Cambridge, UK. ✉e-mail: russell@iau.uni-frankfurt.de; curtius@iau.uni-frankfurt.de

