## [Transparent Peer Review file · Nature Communications]

Isoprene chemistry under upper-tropospheric conditions

Corresponding Author: Mr Douglas Russell

Version 0:

Reviewer comments:

Reviewer #2

(Remarks to the Author)

This manuscript outlines the results from some unique and impactful chamber experiments, presenting analysis that could be important for the future representation of VOC oxidation in global chemical models when considering low pressure and temperature conditions in the upper troposphere that are often overlooked. The highlighted importance of peroxyacetyl nitrate species (aside from the commonly-studied peroxyacetyl nitrate species) in the low-temperature upper troposphere was particularly interesting as a chemical process that is currently excluded from global atmospheric chemistry models. In addition, the use of radical ratios in the analysis as a means to circumvent uncertainties in individual reaction rates was innovative and could provide a valuable methodology for similar investigations in the future. The manuscript was well presented and the methodologies appear to be sound. I recommend the publication of this manuscript, though there are a few additions outlined below that could support the presented arguments.

- Although the authors occasionally mention the potential role of unimolecular RO₂ reactions and RO₂ cross reactions in the chemical system, there are some locations where their impact is not considered:
 - o Lines 30-31: The authors state that multi-stage oxidation is “essential for isoprene to contribute to particle growth”. This fails to consider the potential for HOM formation via unimolecular reactions, or the potential for from C₁₀ accretion products via RO₂ cross reactions which are likely to have a low enough volatility to contribute to aerosol mass, particularly at the low temperatures of interest here. This is noted later in the manuscript at Line 289.
 - o Lines 705-708: The authors state that additional RO₂ from monoterpenes will not impact the system much “especially as the relative distribution in Figure 4 depends solely on HO₂:NO and is independent of RO₂”. It seems unusual to state that RO₂ will not impact the system when this is only true on account of it having been omitted from the analysis. Presumably, at a high enough monoterpene concentration, there would be competition for RO₂ radicals from the new monoterpene RO₂, that would differentially impact the NO, NO₂, and HO₂ reactions based on their relative rates, thus skewing the lines displayed in Figure 4? If the argument is that any monoterpenes (and other VOCs) would be present in low concentrations, so will have little impact on the analysis, then perhaps this phrase could be reworded or removed to avoid confusion.
- Line 221: The authors state that the unsaturated group corresponds to species which have only undergone one OH oxidation. This doesn't seem to be the case since the unsaturated group appears to include IEPOX, which requires two OH oxidation steps. Furthermore, the authors acknowledge in the caption to Figure 2 that “Within this oxidation process, unsaturated products are also produced”, suggesting that this statement within the main text is not strictly correct.
- Lines 753-754: The authors state that their calculated value constitutes an “improvement on current kinetic extrapolations”. While it is understandable that extrapolating the rate data in this way is potentially undesirable, the author's inability to attribute the different calculated ratio value to the NO rate or the HO₂ rate means that their results are highlighting a potential issue with the current representation, rather than providing an improvement. I believe that this contribution is much more carefully described in Lines 491-494 where the authors instead state that the differing results are “...indicating an improvement could be made on the low-temperature extrapolation...”.
- Figure 5 caption: Given the caption's role in describing the figure, rather than providing analysis, the final phrase in this caption regarding the measurement agreement should be confined to the discussion in the main text.
- Line 673, Figure S14: Given the strong dependence of the NO₂/NO ratio on photolysis, it would be potentially valuable to see the seasonal variability, or to at least include the approximate summer and winter time values in the discussion commencing at Line 673. Regardless of whether this additional analysis is provided, the authors should reiterate in this

discussion that the presented values are annual averages, and so values at mid and extreme latitudes could be much lower/higher at any given time of year.

- ‘Laboratory and ambient comparison’ Section: The authors do not include a comparison for the NO₂:NO or HO₂:NO₂ parameters because of “high photolysis of NO₂”. Presumably some NO₂ was still able to be measured during these periods. If so, it would be useful to see the comparisons to the other ratios for a complete analysis, particularly given the ability to compare to later modelled results. Alternatively, if NO₂ was below measurement limits, then this should be stated in the manuscript.

Reviewer #3

(Remarks to the Author)
Review of Russell et al. 2025

Russell et al (2025) present experimental work performed at low temperatures at the CLOUD facility and identify the formation of isoprene oxidation products with large numbers of heteroatoms, specifically those which have undergone two rounds of reaction with OH and then NO, HO₂ or NO₂. These highly oxidised species are likely to have low saturation vapor pressures and are thus more likely to be able to engage in the nucleation of new particles in the low temperatures of the upper troposphere, with attendant impacts on aerosol number and size distributions, CCN and radiative transfer.

The chief novelty of this study centers on the identification of the peroxy nitrate products (i.e. species with the -O₂NO₂ group). These have typically received less attention in the development of isoprene chemical mechanisms given that it has hitherto been assumed that most isoprene chemistry occurs in warm tropical boundary layers where thermal decomposition of the peroxy nitrate is rapid. However, if isoprene can reach the lower temperatures of the upper tropical troposphere, the lifetime of peroxy nitrates with respect to thermal decomposition increases substantially, thus allowing these species to reach concentrations high enough for their nucleation to be appreciable and compete with the less abundant but more involatile hydroperoxide species.

Overall, this study is well presented, interesting and represents an advancement to the understanding of isoprene chemistry and isoprene’s role in the Earth System. However, I have some comments which the authors should address before this paper is considered for publication.

Specific Comments

Figure 1 is well designed but I would argue the production of IEPOX and its subsequent oxidation to produce IP-OOM (lines 155-159) should also be included. The authors should also quote estimated ranges for the saturation vapor pressure for the different oxidation products so that readers can understand the importance of the balance between abundance and C* referred to later in the text. These C* ranges should also reflect the fact that isomers can have different saturation vapor pressures due to the balance between inter- and intra-molecular interactions.

The description of the modelling simulations is lacking, particularly in terms of the new chemistry (i.e. that in Figure 1) given the central role it plays in this study. It is not clear if the new chemistry of Figure 1 is included in the chemistry used to produce the data in Figure 6. This should be clarified and if no additions have been made, this should be stated clearly.

I suggest the authors present a vertical profile of isoprene and its oxidation products, including IP-OOM, from the existing model runs over the Amazon, ideally for the three time points considered so that readers can gain an idea of the concentrations reaching the low temperature regions. It would also help readers to how the standard temperature profile of the atmosphere so they can appreciate the region where this chemistry is valid.

The nucleation rate is directly linked to the saturation vapor pressure of a species. How valid is this?

I would encourage you to include findings of S9 and S10 in the main text (e.g. as part of Figure 4 or 5) since saturation ratio of these compounds is critical to their impact. If the saturation ratio at 243 K does not exceed 1, this should be made clear.

Figure 6(e-g) is a good way to visualize the balance between the HO₂ and NO reaction pathways although I note the NO₂ pathway is missing. Could you include a bar plot showing the relative predicted fluxes through HO₂, NO and NO₂ (over two generations oxidation – 9 possibilities in total) over the Amazon for the three periods of interest? This would give readers a better idea of the expected relative abundance of each species.

In the conclusions, the ratios of k’NO, k’NO₂ and k’HO₂ should be given in an “a:b:c” format so that the relative importance of each pathway is more easily communicated. Likewise, around line 469, use of different denominators makes it harder for readers to understand the relative importance of the pathways.

Version 1:

Reviewer comments:

Reviewer #3

(Remarks to the Author)

The authors have provided detailed and thoughtful responses to my comments and I am happy for the manuscript to be published. New Figure S13 is a good addition.

REVIEWER COMMENTS

1st July 2025

Nature Communications manuscript NCOMMS-25-32955

Title: "Isoprene chemistry under upper tropospheric conditions"

First author: Russell, Douglas

We would like to thank the two reviewers for their support, and their helpful comments that have improved the manuscript. We have edited the manuscript in order to align with a Nature communications article as well as revised the text and display items in response to comments made by the reviewers. Reviewer comments are copied below in black Calibri font and our point-by-point responses are below in blue Times New Roman font. Our replies refer to line numbers in the revised manuscript and any modifications to the text are presented in italics.

Reviewer #2 (Remarks to the Author):

This manuscript outlines the results from some unique and impactful chamber experiments, presenting analysis that could be important for the future representation of VOC oxidation in global chemical models when considering low pressure and temperature conditions in the upper troposphere that are often overlooked. The highlighted importance of peroxyacetyl species (aside from the commonly-studied peroxyacetyl species) in the low-temperature upper troposphere was particularly interesting as a chemical process that is currently excluded from global atmospheric chemistry models. In addition, the use of radical ratios in the analysis as a means to circumvent uncertainties in individual reaction rates was innovative and could provide a valuable methodology for similar investigations in the future. The manuscript was well presented and the methodologies appear to be sound. I recommend the publication of this manuscript, though there are a few additions outlined below that could support the presented arguments.

We would like to thank the reviewer for highlighting the novelty and significance of our study towards global chemistry models as well as recognition of a new methodology for radical investigations. We also thank the reviewer for the recommendation of publication in Nature Communications.

- Although the authors occasionally mention the potential role of unimolecular RO₂ reactions and RO₂ cross reactions in the chemical system, there are some locations where their impact is not considered:

- o Lines 30-31: The authors state that multi-stage oxidation is "essential for isoprene to contribute to particle growth". This fails to consider the potential for HOM formation via unimolecular reactions, or the potential for from C₁₀ accretion products via RO₂ cross reactions which are likely to have a low enough volatility to contribute to aerosol mass, particularly at the low temperatures of interest here. This is noted later in the manuscript at Line 289.

We agree with the reviewer that unimolecular HOM and accretion product formation would occur and form HOM at boundary layer conditions, therefore, to highlight the role of these processes we have revised the text as follows:

Line 31-33: "*Hence, under boundary-layer temperatures, secondary oxidation, autoxidation and C₁₀ accretion product formation via RO₂ cross-reactions are essential for ...*"

We also agree that the lower volatility C10 accretion products would also contribute to aerosol formation at lower temperatures, and Shen et al. 2024 [28] confirms that total IP0N and total IP-OOM is required to constrain nucleation and aerosol growth, respectively, which includes the contribution from C10 accretion products. Note that this study focuses on the formation processes and distribution of the major highly oxidised gas phase products, therefore, we focus on the larger concentration of C5 monomers.

In order to highlight this we have adapted the text as follows:

Line 584-589: *“Importantly, Shen et al.[28] showed that the driver of the formation and growth of new particles in the upper troposphere requires total IP0N and IP-OOM, respectively, including the C10 accretion products. Therefore, it is likely a ...”*

Line 589 – 590: *“... nucleating non-nitrates are contributing, all enhanced ...”*

o Lines 705-708: The authors state that additional RO2 from monoterpenes will not impact the system much “especially as the relative distribution in Figure 4 depends solely on HO2:NO and is independent of RO2”. It seems unusual to state that RO2 will not impact the system when this is only true on account of it having been omitted from the analysis. Presumably, at a high enough monoterpene concentration, there would be competition for RO2 radicals from the new monoterpene RO2, that would differentially impact the NO, NO2, and HO2 reactions based on their relative rates, thus skewing the lines displayed in Figure 4? If the argument is that any monoterpenes (and other VOCs) would be present in low concentrations, so will have little impact on the analysis, then perhaps this phrase could be reworded or removed to avoid confusion.

We agree with the reviewer that at high monoterpene (MT) concentrations, MT RO2 would compete with IP RO2 for NO, NO2 and HO2. And due to differing reaction rate coefficients, this would alter the atmospheric radical ratios observed, indicating a shift in the position of the atmospheric system along the x axis in Figure 4. However, we believe that the shape of lines in Figure 4 would remain the same, as the presence of an MT RO2 would not affect the chemical reaction of IP RO2 + NO, + HO2 or + NO2, only the absolute radical ratios.

To clarify this we have added:

Line 731-735: *“isoprene chemical system will be relatively unaffected, however, elevated monoterpene RO2 concentrations would alter atmospheric radical ratios due to differing reactivities of monoterpene RO2 with NO, HO2, and NO2 compared to isoprene RO2.”*

Additionally, we also agree with the reviewer that under these conditions any monoterpenes / other VOCs would be present in low concentrations, roughly 10% of isoprene concentrations. More importantly, due to the fast reactivity of isoprene with OH compared to other VOCs ($k(\text{isoprene} + \text{OH}) \sim 1.44\text{e-}10$, $k(\text{alpha-pinene} + \text{OH}) \sim 8.4\text{e-}11$ at 223K, IUPAC Aeris) the concentration of isoprene RO2 would far exceed that of MT during the day, therefore, leaving the isoprene system relatively unaffected.

- Line 221: The authors state that the unsaturated group corresponds to species which have only undergone one OH oxidation. This doesn't seem to be the case since the unsaturated group appears to include IEPOX, which requires two OH oxidation steps. Furthermore, the authors acknowledge in the caption to Figure 2 that “Within this oxidation process, unsaturated products are also produced”, suggesting that this statement within the main text is not strictly correct.

The reviewer is correct that the unsaturated group does include species, which have undergone more than one OH oxidation. Although IEPOX is not included in the unsaturated group as it does not classify as an IP-OOM (because it only has three oxygen atoms), oxidation products of IEPOX are

included in IP-OOM as well as other carbonyls, which have likely undergone more than one OH oxidation.

Therefore, the description of this group has been clarified:

Line 224-227: *“The second group (Unsaturated) with an average H:C of 1.89, includes a combination of first and second generation unsaturated molecules which contain ...”*

- Lines 753-754: The authors state that their calculated value constitutes an “improvement on current kinetic extrapolations”. While it is understandable that extrapolating the rate data in this way is potentially undesirable, the author’s inability to attribute the different calculated ratio value to the NO rate or the HO₂ rate means that their results are highlighting a potential issue with the current representation, rather than providing an improvement. I believe that this contribution is much more carefully described in Lines 491-494 where the authors instead state that the differing results are “...indicating an improvement could be made on the low-temperature extrapolation...”.

We agree with the reviewer and have reflected the description in line 491-494 in the conclusion below:

Line 777-780: *“ratio of effective reaction rate coefficients, $k'_{NO_2}:k'_{HO_2}:k'_{NO}$ to be 0.3:2.5:1 at 223K; providing radical reactivity ratios under upper-tropospheric conditions and highlighting an improvement could be made to current kinetic extrapolations.”*

Although, we do not state a direct improvement on kinetic rate coefficients, we believe the ratio provided in this manuscript will be extremely useful in combination with current kinetic extrapolations to build a clear picture of the atmospheric processes.

- Figure 5 caption: Given the caption’s role in describing the figure, rather than providing analysis, the final phrase in this caption regarding the measurement agreement should be confined to the discussion in the main text.

The final sentence has been removed from the figure caption.

And Line 524 now reads *“fall in good agreement with our results.”*

- Line 673, Figure S14: Given the strong dependence of the NO₂/NO ratio on photolysis, it would be potentially valuable to see the seasonal variability, or to at least include the approximate summer and winter time values in the discussion commencing at Line 673. Regardless of whether this additional analysis is provided, the authors should reiterate in this discussion that the presented values are annual averages, and so values at mid and extreme latitudes could be much lower/higher at any given time of year.

We agree with the reviewer that seasonal changes would lead to changes in the NO₂:NO ratio due to photolysis. Looking at seasonal averages, we see the values of NO₂:NO for sunrise, midday, sunset are as follows.

DJF: min = 0.16,0.11,0.16, max = 3.1,1.0,2.1; MAM: min = 0.16,0.12,0.15, max = 1.3,0.9,1.3; JJA: min = 0.11,0.12,0.16, max = 2.8,1.4,2.3; SON: min = 0.17,0.12,0.16, max = 1.6,1.1,1.4.

Minimum values do not change so dramatically, as there are minimal seasonal changes around the equator, however, at higher latitudes there are significant changes in value and location, this model gives read outs up to 51 degrees latitude, beyond these values would continue to increase / decrease further beyond these values depending on the season. As a result we have emphasised the presented

values are annual averages and seasonal and daily values could give much higher / lower throughout the year:

Line 697-702: *“the annually averaged NO₂:NO ranges between 0.1-0.4 in the tropics and up to ~1 at mid-latitudes (see Figure S14). However, this ratio would be subject to a high seasonal variability, therefore, values at higher latitudes could be much larger or smaller dependent on the time of the year.”*

We agree with the reviewer that there would be higher/lower values at high latitudes. However, we think that the addition of these values does not add clarity to the reader, as the majority of high isoprene concentrations in the upper troposphere are likely located around the equator due to the more frequent occurrence of deep convective systems on average throughout the year. At the equator, the NO₂:NO ratio is fairly constant throughout the year due to lower seasonal variations.

- ‘Laboratory and ambient comparison’ Section: The authors do not include a comparison for the NO₂:NO or HO₂:NO₂ parameters because of “high photolysis of NO₂”. Presumably some NO₂ was still able to be measured during these periods. If so, it would be useful to see the comparisons to the other ratios for a complete analysis, particularly given the ability to compare to later modelled results. Alternatively, if NO₂ was below measurement limits, then this should be stated in the manuscript.

The reviewer is correct, although there is a high photolysis of NO₂ during the day, there is still NO₂ present and quantifiable. For the same periods the NO₂:NO ratio ranges between 0.7-2.1, therefore, corresponding to only a small fraction of signal going towards diperoxynitrate, C₅H₁₀N₂O₁₀, from Figure 4. It is important to note also that NO₂ measurement on the aircraft includes thermal artifacts and the nitrate CIMS on the aircraft was experiencing large temperature gradients, therefore, many diperoxynitrates may have thermally decomposed in their inlet. For this reason, a similar plot as Figure 5 for NO₂:NO is not presented, as the double peroxy compound C₅H₁₀N₂O₁₀ was below the limit of detection of the nitrate CIMS, and could not be quantified.

In order to provide atmospheric values from the Extended Data Table 1 in Curtius et al. [25] for comparison to the later model results, we have added the following:

Line 540-544: *“NO₂:NO ratios from the same flight, range between 0.7-2.1 (see ED Table 1 Curtius et al.), placing the system in a nitrate dominated regime – it is important to note that the NO₂ measurement on the aircraft includes thermal artifacts from peroxy nitrates.”*

Reviewer #3 (Remarks to the Author):

Review of Russell et al. 2025

Russell et al (2025) present experimental work performed at low temperatures at the CLOUD facility and identify the formation of isoprene oxidation products with large numbers of heteroatoms, specifically those which have undergone two rounds of reaction with OH and then NO, HO₂ or NO₂. These highly oxidised species are likely to have low saturation vapor pressures and are thus more likely to be able to engage in the nucleation of new particles in the low temperatures of the upper troposphere, with attendant impacts on aerosol number and size distributions, CCN and radiative transfer.

The chief novelty of this study centers on the identification of the peroxy nitrate products (i.e. species with the -O₂NO₂ group). These have typically received less attention in the development of isoprene

chemical mechanisms given that it has hitherto been assumed that most isoprene chemistry occurs in warm tropical boundary layers where thermal decomposition of the peroxy nitrate is rapid. However, if isoprene can reach the lower temperatures of the upper tropical troposphere, the lifetime of peroxy nitrates with respect to thermal decomposition increases substantially, thus allowing these species to reach concentrations high enough for their nucleation to be appreciable and compete with the less abundant but more involatile hydroperoxide species.

We thank the reviewer for their comments highlighting the novelty and potential importance of the peroxy nitrate formation described in our study.

Overall, this study is well presented, interesting and represents an advancement to the understanding of isoprene chemistry and isoprene's role in the Earth System. However, I have some comments which the authors should address before this paper is considered for publication.

We would also like to thank the reviewer for the kind comments and for underlining the importance of this work for understanding isoprene chemistry in the atmosphere.

Specific Comments

Figure 1 is well designed but I would argue the production of IEPOX and its subsequent oxidation to produce IP-OOM (lines 155-159) should also be included. The authors should also quote estimated ranges for the saturation vapor pressure for the different oxidation products so that readers can understand the importance of the balance between abundance and C^* referred to later in the text. These C^* ranges should also reflect the fact that isomers can have different saturation vapor pressures due to the balance between inter- and intra-molecular interactions.

We agree with the reviewer that the production of IEPOX is important, especially at warmer temperatures. However, given that three OH oxidations are required for IEPOX oxidation products to contribute to IP-OOM, and that the rate coefficient for ISOPOOH + OH is approximately 10 times faster than the rate coefficient for IEPOX + OH we expect ISOPOOH, IHN and IHPN to contribute significantly more to IPOOM than IEPOX. Therefore, we would refrain from adding IEPOX to Figure 1 to avoid complication of the key chemical processes. IEPOX formation could occur from IHN or ISOPOOH + OH.

Using group contributions and Clausius-Clapeyron the volatility of IEPOX at 223 K is $8.4 \times 10^{-3} \mu\text{g m}^{-3}$ (D'Ambro et al., 2017 [13]). Whereas, using SIMPOL and Clausius-Clapeyron the volatility of ISOPOOH, an isomer of IEPOX, is $0.042 \mu\text{g m}^{-3}$ indicating that the difference in volatility for the same sum formulae is large. For IHPN* and IHN, SIMPOL and Clausius-Clapeyron give volatilities at 223 K of 0.051 and $0.12 \mu\text{g m}^{-3}$, respectively Pankow et al., 2008 [7].

In comparison, $\text{C}_5\text{H}_{12}\text{O}_6$, $\text{C}_5\text{H}_{10}\text{N}_2\text{O}_8$ and $\text{C}_5\text{H}_{10}\text{N}_2\text{O}_{10}$ have volatilities from SIMPOL and Clausius-Clapeyron of 6.5×10^{-8} , 4.7×10^{-7} , and $7.3 \times 10^{-8} \mu\text{g m}^{-3}$, respectively. However, Kurten et al., 2018 [42] suggests that the variation in saturation vapour pressure could be up to a factor of 100 and 20 for structural and stereoisomers, respectively.

Line 25: *“At boundary layer temperatures, first-generation products, such as isoprene hydroxy hydroperoxide (ISOPOOH, $C^* \sim 5.6 \times 10^4 \mu\text{g m}^{-3}$)”*

Line 37-38: *“(IEPOX, $C^* \sim 1.7 \times 10^4 \mu\text{g m}^{-3}$ [13])”*

Line 553- 556: *“Importantly, these values are average volatilities for all sum formula isomers and variations can be up to a factor of 100 and 20 for structural and stereo-isomers, respectively [42]”*

The description of the modelling simulations is lacking, particularly in terms of the new chemistry (i.e. that in Figure 1) given the central role it plays in this study. It is not clear if the new chemistry of Figure 1 is included in the chemistry used to produce the data in Figure 6. This should be clarified and if no additions have been made, this should be stated clearly.

As the reviewer correctly mentions no new chemistry has been added to the global model, and only the simple MIM chemistry Pöschl et al., 2000 [59] is used. In this study, the model is used to provide radical ratios, which can provide an impression of the global distribution and the implications of the chemistry explored in this manuscript. It is our view that the model results give some additional context to the experimental results, which constitute the core of this study.

A follow up study is planned, which will compare the simple MIM chemistry and the chemistry provided in this manuscript in a global model.

Line 953-958: “... and 300 chemical reactions – the proposed chemistry described in this study has not been included in the simulation. This study uses the simulated atmospheric radical ratios to provide a global implication of the chemistry explored here; a follow up study is planned on modelling the full implementation of this low temperature chemistry.”

Additionally, the global model data used in this study are from the same simulation as used in Tripathi et al., 2025 [46] In this study a detailed description of the model is provided. To highlight this, we have added a reference to this study.

Line 967: “This study uses the same simulation described in Tripathi et al.; we simulated the year 2022 ...”

I suggest the authors present a vertical profile of isoprene and its oxidation products, including IP-OOM, from the existing model runs over the Amazon, ideally for the three time points considered so that readers can gain an idea of the concentrations reaching the low temperature regions. It would also help readers to know the standard temperature profile of the atmosphere so they can appreciate the region where this chemistry is valid.

We agree with the reviewer that the isoprene vertical profile for the three time periods is a useful addition to highlight the isoprene concentrations in low temperature regions. Tripathi et al., 2025 [46] Supplementary Figure 8, presents an isoprene vertical profile comparing model and observations, Tripathi et al. also shows observations of isoprene and isoprene oxidation products. For our study we have added the annual and seasonal variation of the isoprene vertical profile.

We have added figure S11 to the Supplementary Information:

“Figure S11: Isoprene vertical profile above the ATTO tower. An annually averaged isoprene vertical profile (black) of the column above the ATTO tower (latitude = [-1.865,-3.731], longitude = [-57.1875, -59.0625]) for the three time periods sunrise (solid), midday (dashed), and sunset (dot-dashed). Additionally, the shaded areas represent the seasonally averaged isoprene vertical profiles: MAM (orange), JJA (red), SON (beige), DJF (blue). These are shaded between minimal and maximal values for sunrise, midday and sunset.”

And line 615: “dry season, see Figure S11.”

However, as the simulation is using the MIM chemistry mechanism and some chemistry pathways described in this study are not included, we think that a vertical profile of the model’s IP-OOM output would be potentially misleading for the reader.

We thank the reviewer for the suggestion of adding a standard temperature profile. We think this is a good clarification to understand the extent of the proposed chemistry and it helps to guide the reader to regions where the low temperature chemistry explored here is important. Therefore, we have added figure S14:

“Figure S14: Standard temperature profile. The annually averaged global model output temperature as a function of latitude and altitude. Further details on the simulation can be found in the Methods section.”

Reference to the figure is added in the main text as follows:

Line 672-674: “Moreover, the low temperatures explored in this study occur throughout the upper troposphere, see Figure S14, and reach as low as 5 km at higher latitudes.”

The nucleation rate is directly linked to the saturation vapor pressure of a species. How valid is this?

The nucleation rate is directly linked to the saturation ratio, and therefore to the saturation vapour pressure (Seinfeld & Pandis, 2016). The nucleation rate dependence of the gas phase concentration of the substances involved in nucleation of the IP-OOM is already presented and discussed in Shen *et al.*, 2024 [28], and, therefore, we refrain from discussing it in our paper. The quantitative relation between nucleation rate and the vapor pressures of the nucleating IP-OOM is a function also of the concentrations of co-nucleating substances such as inorganic acids. However, the gas phase concentration is also very important to a species’ contribution to the nucleation rate. A single species with an ultra-low saturation vapour pressure but also very low concentration causes very low nucleation rates (e.g., $J < 1 \times 10^{-3} \text{ cm}^{-3} \text{ s}^{-1}$) even if it nucleates at the kinetic collision limit, because the collision frequency of the nucleating molecules is very low. In this case the competing loss processes such as condensation of nucleating molecules and clusters on pre-existing particles or reactor walls will further reduce the observed nucleation rates (Ehrhart and Curtius, 2013).

Seinfeld, J. H., & Pandis, S. N. (2016). *Atmospheric chemistry and physics: From air pollution to climate change* (3rd ed.). Wiley. ISBN: 978-1-118-94740-1

Ehrhart, S. and Curtius, J.: Influence of aerosol lifetime on the interpretation of nucleation experiments with respect to the first nucleation theorem, *Atmos. Chem. Phys.*, 13, 11465–11471, <https://doi.org/10.5194/acp-13-11465-2013>, 2013.

I would encourage you to include findings of S9 and S10 in the main text (e.g. as part of Figure 4 or 5) since saturation ratio of these compounds is critical to their impact. If the saturation ratio at 243 K does not exceed 1, this should be made clear.

We appreciate the suggestion from the reviewer to include the findings of S9 and S10 in the main text and would like to direct to the discussion from Line 543 to 580, which explains the role of saturation ratio and the impact it has for these compounds.

We would like to refrain from including S9 and S10 in Figure 4 or 5. The main aim of this study is to understand the isoprene chemistry at low temperatures. While volatility is important towards particle formation, this study does not focus on nucleation and growth but rather the chemical distribution and formation of these products. At the same time, Figure 4 and 5 would be over-burdened if we include S9 and S10 in them. Additionally, we are planning a separate study, which will further address the volatility and role of these species in nucleation.

For the 243 K experiments, the total saturation ratio from C₅H₁₂O₆, C₅H₁₁NO₇ and C₅H₁₀N₂O₈ is much smaller than for the 223 K experiments shown in Figure S10. Nevertheless, it is always above 1 for the experiments. The lowest value is ~2 at a HO₂:NO of 2x10⁻³, up to approximately 580 at HO₂:NO of ~60. For comparison, the lowest 223 K data point is at a saturation ratio > 1100 and the highest saturation ratio is greater than 3.7x10⁴.

The 243 K data also shows an increase in the combined saturation ratio of these compounds with HO₂:NO. This is not included in Figure S10 because we do not have enough data points at this temperature.

Figure 6(e-g) is a good way to visualize the balance between the HO₂ and NO reaction pathways although I note the NO₂ pathway is missing. Could you include a bar plot showing the relative predicted fluxes through HO₂, NO and NO₂ (over two generations oxidation – 9 possibilities in total) over the Amazon for the three periods of interest? This would give readers a better idea of the expected relative abundance of each species.

We agree with the reviewer and would like to thank them for the suggestion. We have added a figure describing the relative predicted fluxes from isoprene to the six termination products. A description of the calculations has also been added to the Supplementary Information.

Figure S13: Relative predicted flux using relative reaction rate coefficients derived from Figure 4. Relative predicted flux, defined in SI sect. H, describes the predicted formation of each second generation species for sunrise (beige), midday (orange), and sunset (red) using the relative reaction rate coefficients derived from Figure 4. The nine species represent the nine different formation mechanisms depicted in Figure 1. For cross products (C₅H₁₁NO₇, C₅H₁₁NO₈ and C₅H₁₀N₂O₉) the first generation intermediate is put in brackets: (N) is IHN, (P) is IHPN and (H) is ISOPOOH. Each

bar is normalised to the total midday flux to give an indication of the total organic formation and relative branching.”

Additionally, we have added the following sentences to the main text to reflect the results.

Line 660-664: “The relative predicted fluxes clearly depict this in Figure S13. The total product formation increases at midday as explained in Figure S12. Simultaneously, the relative distribution changes from C₅H₁₀N₂O₈-dominated to C₅H₁₁NO₇-dominated.”

An explanation of the term relative predicted fluxes has been added as a new subsection to the supplementary:

“SI sect. H: Predicted Flux

The mechanism depicted in Figure 1 starts from isoprene and creates six distinct products through nine pathways. The flux of each pathway can be used to compare the weight and branching of each pathway under the scenarios created for Figure 6 (b-g).

$$\text{Flux}(i) = (k_{IP} \cdot [\text{OH}'][\text{IP}]) \cdot (k_X[\text{X}]) \cdot (k_Y \cdot [\text{OH}']) \cdot (k_Z[\text{Z}]) \quad (\text{S17})$$

Where k_{IP} , k_X , k_Y , and k_Z are the reaction rate coefficients for Isoprene + OH, ISOPOO + A, first generation product (Y) Y + OH and the second generation RO₂ (Z) Z + A. A is one of HO₂, NO and NO₂

However, as explained above there are uncertainties associated with reaction rate coefficients, k_X and k_Z . Instead relative reaction rate coefficients can be substituted in by dividing the whole equation by $(k_{NO}[\text{NO}'])^2$ to account for two sets of termination. This transforms Equation S17 into Equation S18, Equation S19, Equation S20, and Equation S21 for C₅H₁₀N₂O₈, C₅H₁₁NO₇(N), C₅H₁₂O₆ and C₅H₁₁NO₈(P), respectively.

$$\text{Relative Flux}(C_5H_{10}N_2O_8) = (k_{IP} \cdot [\text{OH}'][\text{IP}]) \cdot (k_{IHN} \cdot [\text{OH}']) = k_{IP}k_{IHN} \cdot [\text{OH}']^2[\text{IP}] \quad (\text{S18})$$

$$\begin{aligned} \text{Relative Flux}(C_5H_{11}NO_7(N)) &= \frac{(k_{IP} \cdot [\text{OH}'][\text{IP}]) \cdot (k_{IHN} \cdot [\text{OH}']) \cdot (k_{HO_2}[\text{HO}_2'])}{(k_{NO}[\text{NO}'])} \\ &= k_{IP}k_{IHN}[\text{OH}']^2[\text{IP}] \frac{k_{HO_2}[\text{HO}_2']}{k_{NO}[\text{NO}']} \\ &= k_{IP}k_{IHN}[\text{OH}']^2[\text{IP}] \cdot T_{HO_2} \frac{[\text{HO}_2']}{[\text{NO}']} \end{aligned} \quad (\text{S19})$$

$$\begin{aligned} \text{Relative Flux}(C_5H_{12}O_6) &= \frac{(k_{IP} \cdot [\text{OH}'][\text{IP}]) \cdot (k_{HO_2}[\text{HO}_2']) \cdot (k_{ISOPOOH} \cdot [\text{OH}']) \cdot (k_{HO_2}[\text{HO}_2'])}{(k_{NO}[\text{NO}'])^2} \\ &= k_{IP}k_{ISOPOOH}[\text{OH}']^2[\text{IP}]k_{HO_2}[\text{HO}_2'] \frac{(k_{HO_2}[\text{HO}_2'])^2}{(k_{NO}[\text{NO}'])^2} \\ &= k_{IP}k_{ISOPOOH}[\text{OH}']^2[\text{IP}] \cdot T_{HO_2}^2 \frac{[\text{HO}_2']^2}{[\text{NO}']^2} \end{aligned} \quad (\text{S20})$$

$$\begin{aligned} \text{Relative Flux}(C_5H_{11}NO_8(P)) &= \frac{(k_{IP} \cdot [\text{OH}'][\text{IP}]) \cdot (k_{NO_2}[\text{NO}_2']) \cdot (k_{IHPN} \cdot [\text{OH}']) \cdot (k_{HO_2}[\text{HO}_2'])}{(k_{NO}[\text{NO}'])^2} \\ &= k_{IP}k_{IHPN}[\text{OH}']^2[\text{IP}]k_{HO_2}[\text{HO}_2'] \frac{(k_{HO_2}k_{NO_2}[\text{HO}_2'][\text{NO}_2'])}{(k_{NO}[\text{NO}'])^2} \\ &= k_{IP}k_{ISOPOOH}[\text{OH}']^2[\text{IP}] \cdot T_{HO_2} T_{NO_2} \frac{[\text{HO}_2'][\text{NO}_2']}{[\text{NO}']^2} \end{aligned} \quad (\text{S21})$$

Here we use known reaction rate coefficients for k_{IP} and $k_{ISOPOOH}$ in combination with reaction rate ratios from Figure 4 and concentrations taken from the global model.

Concentrations correspond to the mean output concentrations from the global model at an isothermal surface of 223 K above the Amazon (latitude = [20,-40], longitude = [-81,-30]) for sunrise, midday and sunset.

The relative predicted fluxes can be calculated for each term. Normalising by the sum of all fluxes for the midday section allows for a direct comparison in Figure S13.”

In the conclusions, the ratios of $k'NO$, $k'NO_2$ and $k'HO_2$ should be given in an “a:b:c” format so that the relative importance of each pathway is more easily communicated. Likewise, around line 469, use of different denominators makes it harder for readers to understand the relative importance of the pathways.

Line 475: added into the equation for clarity is “ $k'NO_2:k'HO_2:k'NO \sim 0.3:2.5:1$ ”

Line 776-780: “*ratio of effective reaction rate coefficients, $k'NO_2:k'HO_2:k'NO$ to be 0.3:2.5:1 at 223K; providing radical reactivity ratios under upper-tropospheric conditions and highlighting an improvement could be made to current kinetic extrapolations.*”

Additional Comments

Tripathi et al. has been published in Nature Communications, therefore, the citation has been amended to reflect this.

The $k(\text{Isoprene} + \text{OH})$ rate coefficient used for Figure S1 has been updated. Originally it was using a value from IUPAC of $2.1 \times 10^{-11} \exp(465/T)$, however, we have now changed to the expression quoted in Wennberg et al. (IUPAC) $2.7 \times 10^{-11} \exp(390/T)$ in order to be consistent with the rate coefficient used in Figure 6. For temperatures of 300 and 223 K the value changes from 9.89×10^{-11} to 9.91×10^{-11} and 1.69×10^{-10} to 1.55×10^{-10} . This change only affects Figure S1 and does not change the conclusions or picture of the figure. This includes changing the caption.

Abstract: “*Isoprene (C5H8) is the non-methane hydrocarbon with the highest emissions to the atmosphere.*”

Inline fractions have been expressed as ratios rather than fractions therefore the following corrections have been made:

Line 196: kNO_2/kNO has been changed to $kNO_2:kNO$

Line 198: kNO_2/kNO has been changed to $kNO_2:kNO$

Line 459: $[HO_2]/[NO] \sim k'NO/k'HO_2$ has been changed to “ $[HO_2]:[NO] \sim k'NO:k'HO_2$ ”

Line 464: $k'NO/k'HO_2$ has been changed to “ $k'NO:k'HO_2$ ”

Additionally, the colouring of lines in Figures 4, 5 and S8 have been changed to force the pink points (now purple) to stand out more. This does not change the conclusions or picture of the figure. The captions have been changed to reflect this.

Figure 4 caption: “... *the mixed product C5H10N2O9 (purple), and the ...*”

“... *products C5H10N2O8 (blue), C5H11NO7 (purple) and C5H12O6 (red), and (c) presents C5H10N2O10 (blue), C5H11NO8 (purple) and C5H12O6 (red) as...*”

Figure 5 caption: “... *experimental fits of C5H10N2O8 (blue), C5H11NO7 (purple) and C5H12O6 (red) from ...*”

Figure S8 captions: “...*and the aircraft (213 K and 187 mbar) [20] for C5H10N2O8 (blue), C5H11NO7 (purple), and C5H12O6 (red).*”

REVIEWER COMMENTS

11th August 2025

Nature Communications manuscript NCOMMS-25-32955A

Title: "Isoprene chemistry under upper tropospheric conditions"

First author: Russell, Douglas

Reviewer #3 (Remarks to the Author):

The authors have provided detailed and thoughtful responses to my comments and I am happy for the manuscript to be published. New Figure S13 is a good addition.

We would like to thank the reviewer for their support and recommendation of publication in Nature Communications.